# Elastic conducting polymer composites in thermoelectric modules

Nara Kim [1], Samuel Lienemann [1], Ioannis Petsagkourakis [1], Desalegn Alemu Mengistie [1,2], Seyoung Kee[3], Thomas Ederth [4], Viktor Gueskine[1], Philippe Leclère [5], Roberto Lazzaroni[5], Xavier Crispin [1✉] & Klas Tybrandt [1✉]

The rapid growth of wearables has created a demand for lightweight, elastic and conformal energy harvesting and storage devices. The conducting polymer poly(3,4-ethylenedioxythiophene) has shown great promise for thermoelectric generators, however, the thick layers of pristine poly(3,4-ethylenedioxythiophene) required for effective energy harvesting are too hard and brittle for seamless integration into wearables. Poly(3,4-ethylenedioxythiophene)-elastomer composites have been developed to improve its mechanical properties, although so far without simultaneously achieving softness, high electrical conductivity, and stretchability. Here we report an aqueously processed poly(3,4-ethylenedioxythiophene)-polyurethane-ionic liquid composite, which combines high conductivity (>140 S cm$^{-1}$) with superior stretchability (>600%), elasticity, and low Young's modulus (<7 MPa). The outstanding performance of this organic nanocomposite is the result of favorable percolation networks on the nano- and micro-scale and the plasticizing effect of the ionic liquid. The elastic thermoelectric material is implemented in the first reported intrinsically stretchable organic thermoelectric module.

[1] Laboratory of Organic Electronics, Department of Science and Technology, Linköping University, 601 74 Norrköping, Sweden. [2] Materials Engineering Department, California Polytechnic State University, 1 Grand Ave, San Luis Obispo, CA 93407, USA. [3] School of Chemical Sciences, The University of Auckland, Auckland 1010, New Zealand. [4] Division of Molecular Physics, Department of Physics, Chemistry and Biology, Linköping University, 581 83 Linköping, Sweden. [5] Laboratory for Chemistry of Novel Materials, Materials Research Institute, University of Mons, B 7000 Mons, Belgium.
✉email: xavier.crispin@liu.se; klas.tybrandt@liu.se

The transformation of electronics into soft and stretchable systems opens up for a wide range of novel applications within wearables for e.g. smart clothing[1], healthcare monitoring[2], and biomedical implants[3]. Thermoelectric (TE) generators that can convert body heat into electricity is an attractive energy-harvesting technology for the realization of self-powered wearable electronics. For energy harvesting in wearables, organic TE materials offer critical advantages over their inorganic counterparts (i.e., bismuth-based alloys), including natural abundance of base elements; light weight; easy processability; non-toxicity; and tunable mechanical properties[4,5]. Among organic TE materials, the conducting polymer (CP) poly(3,4-ethylenedioxythiophene) (PEDOT) is considered to be one of the most promising polymers[6], because of its (i) high electrical conductivity ($\sigma_{dc} > 1000 \, \text{S cm}^{-1}$) achieved by various secondary doping methods to modulate the morphology[7], (ii) relatively high Seebeck coefficient ($S \approx 10-50 \, \mu\text{V K}^{-1}$) originating from a bipolaron network (semi-metallic) that has asymmetric density of states at the Fermi level[8], and (iii) intrinsic low thermal conductivity ($\kappa < 1 \, \text{W m}^{-1} \, \text{K}^{-1}$) due to weak lattice vibrations[9]. Altogether, this leads to a dimensionless figure of merit, $ZT = \sigma_{dc}S^2/\kappa$, as high as 0.42 for PEDOT thin films to date[9]. Moreover, the solution processability of aqueous dispersions of PEDOT charge-balanced with poly(styrene sulfonate) (PSS) enables cost-effective and eco-friendly fabrication of TE generators using high-throughput printing techniques[10]. However, pristine PEDOT:PSS has a Young's modulus of $\approx 500 \, \text{MPa}$ and a maximum tensile strain ($\varepsilon$) of $\approx 5 \, \%$[11,12], making it too hard and brittle to be used in wearable TEs as dynamic human motions often involve strains of $20-80\%$[13]. Therefore, in parallel with the studies to enhance $ZT$, significant efforts have recently been made to increase the intrinsic stretchability of PEDOT:PSS.

One strategy to render PEDOT:PSS stretchable is to add non-volatile plasticizers[14] such as surfactants with polyethylene glycol segments (Zonyl FS-300, Triton X-100)[15,16], polyols (xylitol)[17], and ionic liquids (ILs)[12,18,19]. Plasticizers that contain functional groups with high dielectric constant (e.g. ether, hydroxyl) or ions can be effectively intercalated between the polymer chains of PEDOT:PSS and thereby enhance the polymer segmental motion by increasing the free volume. These plasticizers also act as secondary dopants and enhance the $\sigma_{dc}$ of PEDOT:PSS by inducing favorable morphologies within the films. Recently, extremely high stretchability (>800%) in combination with high $\sigma_{dc}$ (>1000 S cm$^{-1}$) was achieved for a thin PEDOT:PSS film deposited on an elastic styrene–ethylene/butylene–styrene (SEBS) substrate from a PEDOT:PSS dispersion containing a specific IL[18]. Although this approach, which makes PEDOT:PSS plastic and malleable, can generate highly stretchable PEDOT:PSS thin films (<1 μm) when supported by elastomer substrates, these films suffer from irreversible deformation (out-of-plane buckles) under cyclic stretching, substrate-dependent stretchability, and severe degradation of stretchability with increasing film thickness[12,14,15,18,19]. Eventually, free-standing PEDOT:PSS-plasticizer thick films (10–100 μm) are neither very stretchable nor elastic; in other words, they don't return to the original shape when released after stretching[20]. This is problematic for wearable TE generators since they require several 100 μm to mm thick TE legs to effectively harvest the temperature gradient between bottom (heat source) and top (the surrounding) sides in a vertical device configuration[21]. Accordingly, the mechanics of TE generators is dominated by the mechanical property of the thick active materials rather than the substrates.

One approach to achieve high stretchability and elasticity in the bulk form of the material is to embed PEDOT:PSS in an elastomeric matrix. Various methods to make PEDOT- or PEDOT:PSS-elastomer composites and their performances as elastic conductors are summarized in Supplementary Table 1. In comparison to the tedious processes (i.e. post-cast polymerization, purification, freeze-drying, re-dispersing, infiltration, curing, etc.) involved in the use of water-insoluble elastomers[22–25], such as PDMS and polyurethane (PU), the blending of PEDOT:PSS dispersion with water-borne PU (WPU) constitutes an easier and more effective way of making elastic composites while preserving the advantages of water-processability[26–28]. A high stretchability in the free-standing state (fracture strain $\varepsilon_f = 530 \, \%$) as well as a superior elastic recovery of 83% was reported for the PEDOT:PSS-WPU composite blended with reduced graphene oxide and dimethyl sulfoxide (DMSO, a well-known secondary dopant for PEDOT:PSS)[28]. However, to obtain such mechanical properties, a high loading (>94%) of insulating WPU was necessary[27,28]. Although this composite can benefit from the low $\kappa$ of WPU (<0.2 W m$^{-1}$ K$^{-1}$)[29], a small content of conductive PEDOT:PSS, along with nonoptimal percolation between PEDOT:PSS domains in the elastomeric matrix, resulted in a low $\sigma_{dc}$ (<20 S cm$^{-1}$). This tradeoff between electrical conductivity and mechanical properties (maximum strain, softness) holds true for all previously reported PEDOT-elastomer composites. Therefore, new PEDOT composites are needed that simultaneously combine high $\sigma_{dc}$ (>100 S cm$^{-1}$) with good intrinsic mechanical properties ($\varepsilon_f >$ 200%, elastic recovery, Young's modulus < 10 MPa) for wearable applications that require thick active materials, including TE generators, supercapacitors/batteries and actuators.

Herein, we report free-standing elastic highly conductive ($\sigma_{dc} >$ 140 S cm$^{-1}$) and highly stretchable ($\varepsilon_f >$ 600%) PEDOT:PSS-IL-WPU composites obtained via solution casting of aqueous blends. The introduction of IL into the elastic PEDOT:PSS-WPU composite has tremendous effects on both electrical and mechanical properties of the composite, as the composite with IL exhibits a 46-times larger elongation at break, a 35-times lower elastic modulus, and 6 orders of magnitude higher $\sigma_{dc}$ in comparison to the PEDOT:PSS-WPU composite without IL. Based on the thermoelectric properties of the composite, in combination with in-house synthesized gold nanowire interconnects, we develop the first intrinsically stretchable organic thermoelectric module. At 40% tensile strain, the output voltage of the module is unchanged while the output power decreases with the increase in the internal resistance of the module, retaining 48% and 83% of the initial power under parallel and perpendicular strain, respectively.

## Results

**Water-processable elastic conducting polymer composites**. The chemical structures of the three components of the composites are shown in Fig. 1a. The PEDOT:PSS is a polyelectrolyte complex of positively doped conducting PEDOT, and negatively charged insulating PSS. PEDOT:PSS can be dispersed in water as a form of charged gel particles, the outer surface of which is dominated by an excess of PSS polyanions[30]. The electrostatic repulsion between the gel particles prevents agglomeration and thus ensures the stability of the dispersion. ILs are salts that exist in the liquid state at temperature below 100 °C due to the bulky and asymmetric structures of the ions. ILs exhibit favorable solvating properties and act as non-volatile plasticizers for a wide range of polar and non-polar compounds[31,32]. Here, we used three different types of water-miscible ILs with a common 1-ethyl-3-methylimidazolium (EMIM) cation and various anions: ethyl sulfate (ES), tricyanomethanide (TCM), and tetracyanoborate (TCB), which are proven to be effective in controlling the morphology of PEDOT:PSS[10]. The PU elastomer is a copolymer made of alternating isocyanate and polyol groups joined by urethane links. The elastomeric properties originate from the co-existence of hard, immobile segments and soft, mobile segments and their phase segregation. Although the

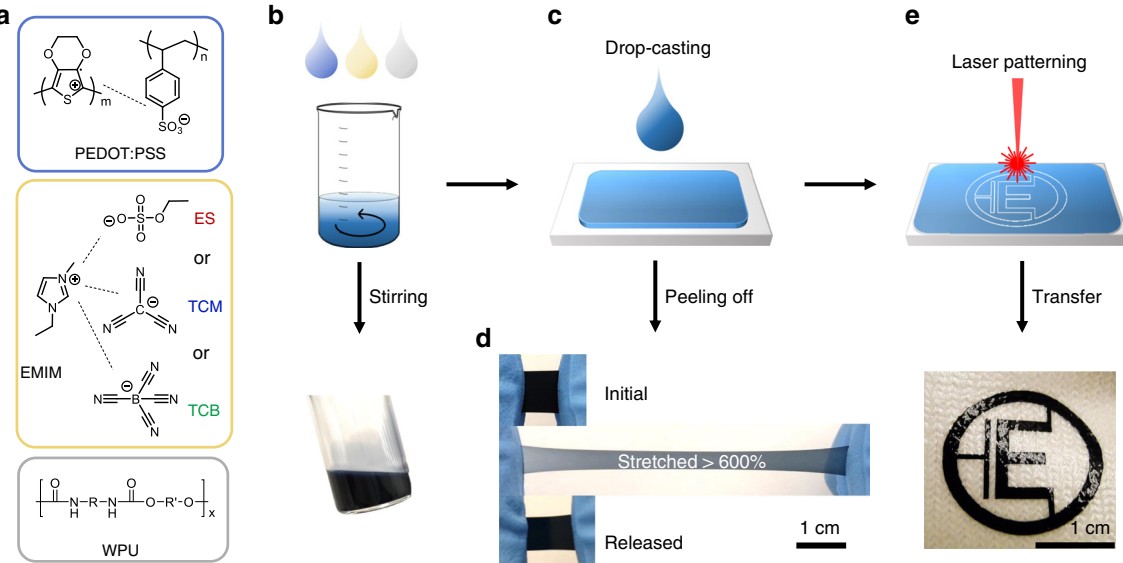

**Fig. 1 Printable elastic conducting polymer composites. a** Chemical structures of PEDOT:PSS (conducting polymer, CP), water-borne polyurethane (WPU), and different types of ionic liquids (ILs), **b** of which aqueous dispersions/solution can be mixed in one pot to formulate the composite ink. **c** The solution processability of the composite ink allows for drop-casting to produce thick films (>40 μm). **d** A free-standing composite film (CP:TCM:WPU = 15:25:85, w/w) can be stretched to over 600% and relax to its original shape with little hysteresis. **e** A laser-patterned composite film can be transferred and adhere to irregular surfaces like textiles.

backbone of PU itself is hydrophobic and insoluble in water, the incorporation of ionic groups or hydrophilic segments enables the dispersion of PU in water, generating WPU[33]. Since all three components are either aqueous dispersions or soluble in water, they can be mixed together to formulate an aqueous ink that enables the production of highly stretchable, elastic and conductive patterns. The ink can be used to coat various substrates or to generate free-standing films via standard coating and printing techniques (Fig. 1b–e).

**Electromechanical and thermoelectric properties.** To elucidate the role of each component in the composite and to find the optimal composition with respect to high conductivity and high stretchability, we investigated the mechanical, electrical, thermoelectric, and electromechanical properties of the composites for a wide range of compositions. Fig. 2a shows stress-strain curves of the free-standing composite films with varying proportion of WPU to PEDOT:PSS (denoted as CP from now on), where the CP dispersion (1.2 wt%) was pre-mixed with 2 wt% of EMIM TCM (i.e. CP:IL = 12:20, w/w) before adding different amounts of WPU. Increasing the proportion of WPU resulted in a higher $\varepsilon_f$ and a lower Young's modulus ($E$) (Fig. 2b). The mechanical properties start to improve rapidly at WPU:CP = 80:20 ($\varepsilon_f \approx 150\%$) and the composite becomes stretchable up to 500% (max. $\varepsilon_f \approx 650\%$) for WPU:CP = 85:15. However, the $\sigma_{dc}$ of the composite gradually decreases with increasing WPU content (Fig. 2c). Nonetheless, $\sigma_{dc}$ still remains above $100\ S\ cm^{-1}$ at WPU:CP = 85:15. Up to this point, the $S$ of the composite is nearly constant ($S \approx 22\ \mu V/K$), but further loading of WPU increased $S$ to 38 μV/K at WPU:CP = 95:5. By considering both conductivity and stretchability, the WPU to CP ratio was optimized to 85:15.

At the optimized CP-WPU composition, the amount of IL has a tremendous effect on the mechanical, electrical, and thermoelectric properties of the composite. Without IL, the composite film fractured under 13% tensile $\varepsilon$, and its $E$ of 235 MPa is in the same order as that of the pristine CP ($\approx$550 MPa). Upon the

addition of the IL, the composite with the same CP-WPU ratio becomes much more stretchable and softer (Fig. 2d, e). The $\varepsilon_f$ increases by 46 times and the $E$ decreases to 6.8 MPa when adding 2 wt% of EMIM TCM into a CP dispersion. More surprisingly, the $\sigma_{dc}$ increases by 6 orders of magnitude (Fig. 2f). Given that the $\sigma_{dc}$ difference between CP and CP-IL is about 3 orders of magnitude[10], the results indicate that IL plays a critical role in the formation of the macroscopic conducting percolative network. As the electrical percolation diminishes with lower amounts of IL, the $S$ increases inversely. Note that the amount of IL has no effect on the $S$ of CP-TCM composites without WPU (see Supplementary Fig. 1). A potential hypothesis to understand the correlation of $S$ with percolation is proposed in Supplementary Fig. 2.

Next we compared the effect of three different ILs on the performance of the composite (Fig. 3a–d). Interestingly, there is no direct relationship between the $\sigma_{dc}$ of the CP-IL-WPU composites (TCM > TCB > ES) and the CP-IL films (TCB > TCM > ES, see Supplementary Fig. 3), meaning that the electrical percolative network in the composites strongly depends on the specific IL. In comparison, the $\sigma_{dc}$ of the composite with DMSO is the lowest, even though the PEDOT:PSS film with DMSO has a $\sigma_{dc}$ over $900\ S\ cm^{-1}$. Also, the mechanical properties of the composites differ with the type of IL but, overall, ILs cause significantly better stretchability and softening effects compared to DMSO. The mechanical property of the CP-DMSO-WPU composite is similar to that of the pristine CP-WPU composite (see Supplementary Fig. 4).

The electromechanical properties were compared by evaluating the relative change in resistance ($R/R_0$) under tensile $\varepsilon$ (Fig. 3b). The increase in resistance caused by geometrical shape changes ($R_G/R_0$, see Supplementary Fig. 5) was plotted to show two regimes: $R/R_0 < R_G/R_0$ indicating the increase in $\sigma_{dc}$, and $R/R_0 > R_G/R_0$ indicating the decrease in $\sigma_{dc}$[34]. The increase in $\sigma_{dc}$ of the composites with ILs under small strain is attributed to the alignment of CP chains or fibrils[18], whereas further stretching over a critical point resulted in a decline in $\sigma_{dc}$ due to the disruption of the electrical network within the composites[15]. For TCM- and ES-added composites, the regime with increase

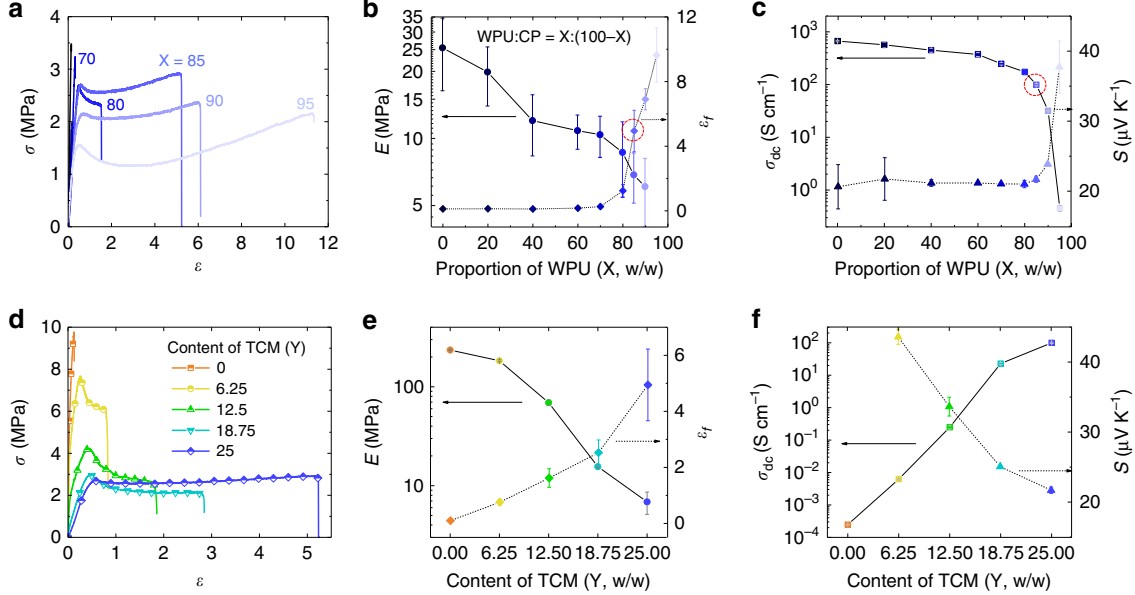

**Fig. 2 Mechanical, electrical, and thermoelectric properties. a** Stress ($\sigma$)-strain ($\varepsilon$) curve, **b** Young's modulus ($E$) and fracture strain ($\varepsilon_f$), **c** electrical dc conductivity ($\sigma_{dc}$) and Seebeck coefficient ($S$) of the composite with varying proportion of WPU ($X$) to CP ($100-X$). The composition ratio of the chosen IL (TCM) to CP was fixed at CP:TCM = 15:25. There is a drastic change in all properties in the range $X = 80-95$. The red dotted circles indicate the data points at the optimized $X$ ($\approx 85$). **d** $\sigma$-$\varepsilon$ curve, **e** $E$ and $\varepsilon_f$, **f** $\sigma_{dc}$ and $S$ of the composite with varying content of TCM ($Y$). The proportion of WPU to CP was fixed at CP:WPU = 15:85. Composition ratio, CP:TCM:WPU = 15:$Y$:85. w/w for all composition ratios. Error bars represent the standard deviation (SD).

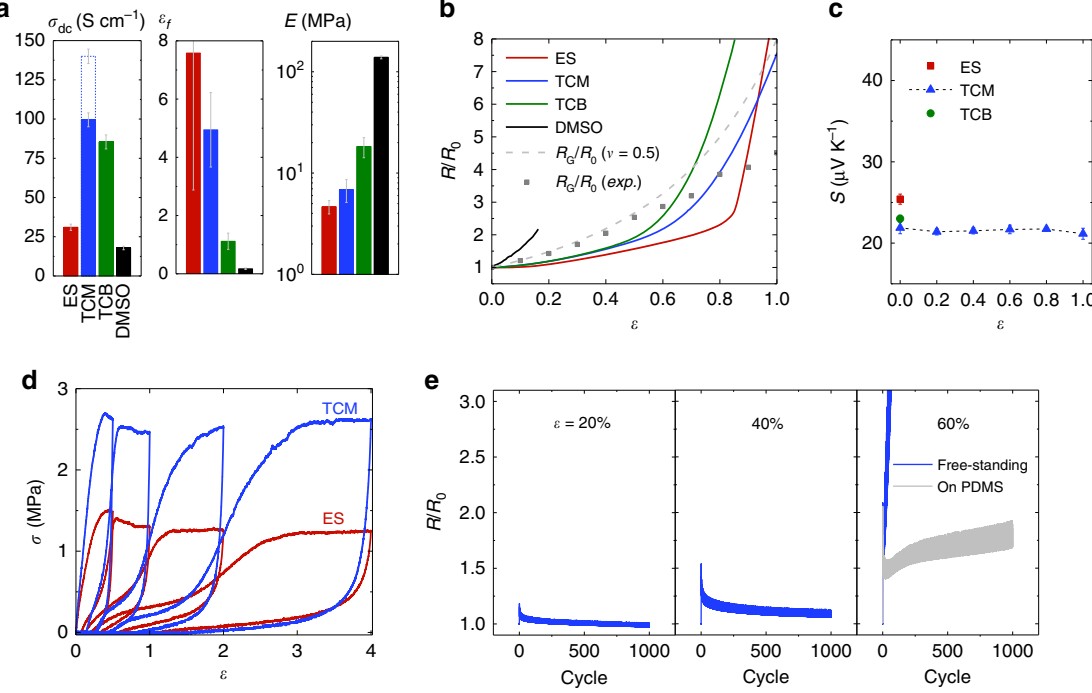

**Fig. 3 Strain dependence of the properties. a** Comparison of $\sigma_{dc}$, $\varepsilon_f$, and $E$ of the composites with a different type of ILs or dimethyl sulfoxide (DMSO). The dotted bar shows the stabilized $\sigma_{dc}$ (see Supplementary Fig. 7). **b** Resistance change ($R/R_0$) upon tensile stretching. Resistance change caused by geometrical shape changes ($R_G/R_0$) was plotted based on the typical Poisson's ratio ($\nu$) for rubber ($\nu = 0.5$) or the experimental data obtained for the composite (see Supplementary Fig. 5). **c** Initial $S$ and $S$ for tensile strain. **d** $\sigma$-$\varepsilon$ curve upon cyclic loading. **e** $R/R_0$ of the optimized composite (CP:TCM:WPU = 15:25:85) upon cyclic tensile stretching (blue lines). $R/R_0$ of the optimized composite supported on a PDMS substrate under cyclic strain at 60% is also shown (gray line). Composition ratio, CP:IL:WPU = 15:25:85. For the composite with DMSO, 5.6 wt% of DMSO was added to the CP dispersion before making the composite. w/w for all composition ratios. Error bars represent the SD.

in $\sigma_{dc}$ is extended to over 80% strain. There seems to be a correlation between good electromechanical performance and low $E$ and high $\varepsilon_f$, as the relative resistance of the softer composites decreases less with $\varepsilon$ (Fig. 3a, b). This can be understood by considering that the well-plasticized PEDOT:PSS phase can deform more readily without fracturing the electrical pathway throughout the composite. $S$ does not vary much depending on the type of ILs, while the inverse relationship between $\sigma_{dc}$ and $S$ of the composite still holds with different ILs that cause different percolative networks (Fig. 3c). The $S$ of the composite with TCM was nearly constant for strains up to 80% and decreased slightly at 100% strain.

The stress-strain curves for the composites with ILs under cyclic loading (Fig. 3d) show hysteresis loops that are characteristic of viscoelastic materials[35], probably due to the mechanical influence of the CP and IL phases. The composites with ES (or TCM) recover ~87% (or 75%) of their original lengths after unloading, respectively, exhibiting decent elastic properties. The $R/R_0$ of the composite with TCM shows excellent cycling stability when stretched up to 40% (Fig. 3e). $R$ became even less than $R_0$ when released after 1000 stretching cycles at 20% or increased by only 6% after 1000 cycles at 40%. The decline in $R$ over cycles is ascribed to the alignment of CP fibrils[18]. However, the viscoelastic property of the composite caused partial plastic deformation at 60% strain and repeated stresses focused on weak points resulted in the complete tear after approximately 200 cycles (see Supplementary Fig. 6). Nonetheless, when supported on a PDMS substrate, more uniformly distributed stresses across the substrate suppress plastic deformation so that the composite could withstand 1000 stretching cycles at 60% with a $R/R_0$ of only 1.7.

The $\sigma_{dc}$ values in Figs. 2 and 3 were based on measurements conducted right after the spin-coated samples were taken out of the vacuum drying chamber. Repeated $\sigma_{dc}$ measurements of the optimized composite (CP:TCM:WPU = 15:25:85) in air revealed that the $\sigma_{dc}$ was increasing over time and stabilized at an average value of 140 S cm$^{-1}$ after 11 h (Fig. 3a and Supplementary Fig. 7b). Since we added a small amount of ammonia (CP:NH$_3$ = 8:1) in all composite dispersions to prevent acid-induced aggregation of the anionic WPU by protonation from PSSH, the initial $\sigma_{dc}$ was lower due to the partial dedoping of the PEDOT occurring in the relatively basic dispersion containing NH$_3$ (see Supplementary Fig. 7a,c)[36,37]. The increase in $\sigma_{dc}$ of the composite over time is attributed to the recovery of the doping level via spontaneous oxidation upon oxygen (air) exposure[38] as evidenced by the changes in absorption spectra (see Supplementary Fig. 7c). Considering the mass fraction of CP in the optimized composite, the $\sigma_{dc}$ of 140 S cm$^{-1}$ translates into an $\sigma_{dc}$ for the CP part of ~1170 S cm$^{-1}$. As the conduction pathways within a composite are often not fully aligned with the macroscopic direction of conduction, the $\sigma_{dc}$ is typically lower in a composite than in a pure conductor[39]. Even for well-connected systems, like meshes of randomly oriented nanowires[40] or connected tubes of 45° angle with respect to the direction of conduction (fiber contact model)[39], the $\sigma_{dc}$ is reduced by 50%. Although the exact geometry of the CP phase in our composite is unknown, by assuming that it possesses the same orientational characteristics as the fiber geometries described above, the effective $\sigma_{dc}$ for the CP phase would be ~2340 S cm$^{-1}$. This is nearly the same value as for the CP film without WPU (≈2350 S cm$^{-1}$, see Supplementary Fig. 3), indicating that most of the PEDOT network within the composite is contributing to the conduction. Compared with the spin-coated film ($\sigma_{dc}$ ≈ 140 S cm$^{-1}$), the drop-cast film of the optimized composite has only a slightly lower $\sigma_{dc}$ (≈100 S cm$^{-1}$), a similar reduction has been observed for CP-DMSO films[41,42]; which indicates that thicker free-standing drop-cast films used below in thermoelectric generators have likely similar transport mechanism and microscopic morphology as spin-coated films.

**Structural modifications by the incorporation of ILs.** To understand the role of ILs in the formation of the outstanding electrical percolation network within the CP-WPU composite while preserving the superior mechanical compliance of WPU, we focused on the structural changes that occur in the dispersion and film states upon addition of ILs. Dynamic light scattering (DLS) measurements show that the pristine CP dispersion has a bimodal size distribution with peaks at 50 nm and 630 nm (Fig. 4a), in agreement with a previous report[43]. Upon the addition of 2 wt% of EMIM TCM, the peak diameter for both populations grows to 170 nm and 3.2 μm, respectively, indicating the strong interaction between the CP and the IL in the dispersion. It has been proven that the counter-ion exchange between PEDOT$^+$PSS$^-$ and EMIM$^+$TCM$^-$ leads to the partial dissociation of PEDOT from its water-soluble template anion, PSS, which causes a molecular rearrangement[10,44]. In contrast, the size distribution of the WPU dispersion remains almost constant after the addition of IL (single peak at 120 nm, Supplementary Fig. 8). When mixed together, the particle size distribution of the CP-WPU composite shows two peaks nearly at the same positions as observed for the CP and WPU dispersions (Fig. 4b). This means that there is no strong interaction between the dispersed particles of the CP and WPU in the mixture. For the CP-IL-WPU composite, the size distribution peak of the CP-IL shifts from 3.2 to 2.1 μm when WPU is added, implying that some interaction between the CP and WPU particles occurs in the presence of IL.

Fig. 4c displays the X-ray diffraction (XRD) patterns of CP films with and without the addition of different types of ILs or DMSO. The pristine CP exhibits several broad diffraction peaks with low intensities. The two characteristic peaks at $2\theta = 6.7°$ and 25.6° correspond to the lamella stacking distance $[d_{(100)}]$ of PEDOT:PSS and the π–π stacking distance $[d_{(010)}]$ between PEDOT rings, respectively (see Supplementary Fig. 9 for molecular packing structure)[45]. When the films were deposited from CP dispersions containing ILs or DMSO, the $d_{(010)}$ peak shifts to a higher angle, indicating a decrease in $d_{(010)}$. For TCM- and TCB-added CP films, the intensity of $d_{(010)}$ peaks increased remarkably, and the full-width at half-maximum (FWHM) of $d_{(010)}$ was reduced from 3.6° to ~2.0°, which translates into an increase in the crystalline domain size along (010)-axis from 22 Å to ~40 Å in the presence of ILs, as estimated with the Scherrer formula[46]. The XRD pattern of WPU shows a large broad peak centered at $2\theta = 19°$, which is a typical feature of amorphous polymers (see Supplementary Fig. 10). Although the change in the diffraction peak of WPU is not significant, the FWHM increases slightly with the addition of ILs, whereas this change is negligible when DMSO was added. This trend becomes more pronounced in the CP-WPU composite: the XRD patterns of the composites in Fig. 4d reveal that an apparent broadening of the amorphous WPU peak, which indicates the reduced domain size of WPU, occurs with the addition of ILs. Diffraction peaks from CP can hardly be distinguished in the patterns of Fig. 4d due to its small proportion in the composite. However, the $d_{(010)}$ and $d_{(100)}$ peaks are clearer for TCM- and TCB-added composites, which have distinctly higher conductivities than the others.

The changes in the morphological properties upon the addition of ILs were investigated by atomic force microscopy (AFM) (Fig. 5). The pristine CP exhibits a granular structure with a grain diameter of approximately 50 nm, which corresponds well to the smaller diameter of the globular gel particles in the CP dispersion. For CP-IL, it was difficult to obtain clear height images, probably due to the presence of excess ILs on the surface of the films, while

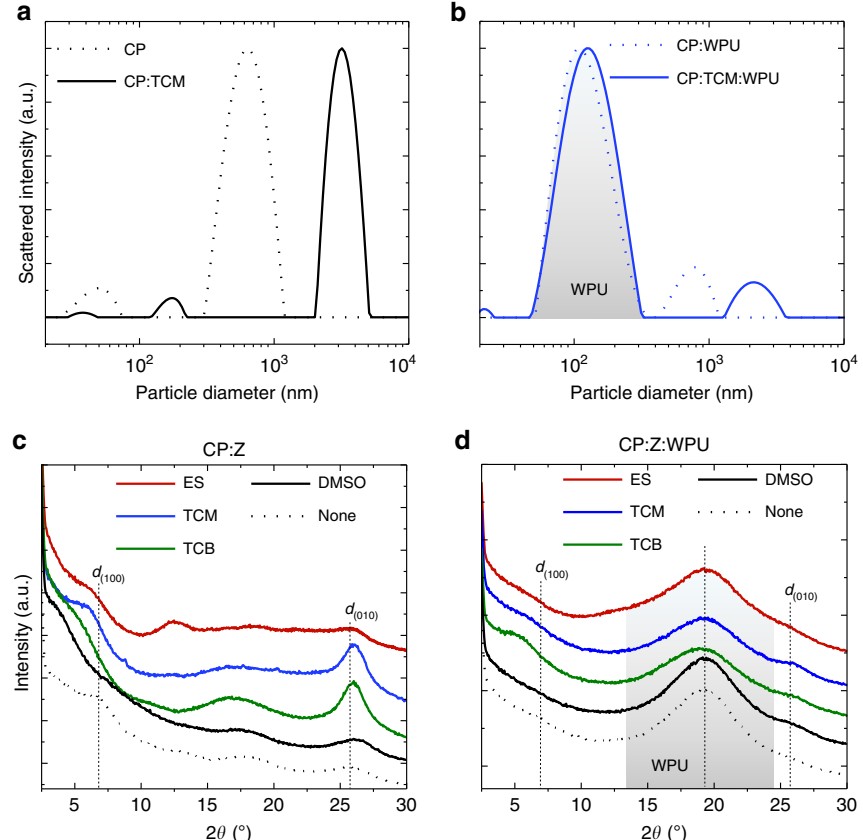

**Fig. 4 Structural changes in the dispersion and film states by the addition of IL.** Dynamic light scattering studies on particle size distribution of **a** CP and **b** CP-WPU composite dispersions, both with and without TCM. The size distribution of WPU was filled with gray color. XRD patterns of **c** CP and **d** CP-WPU composite films, both with and without a different type of ILs or DMSO (Z). The diffraction peak of WPU was filled with gray color. $d_{(100)}$ and $d_{(010)}$ indicate the lamella stacking distance and the π–π stacking distance detected in pristine PEDOT:PSS, respectively. Composition ratio, CP:IL = 15:25 or CP:IL:WPU = 15:25:85. w/w for all samples.

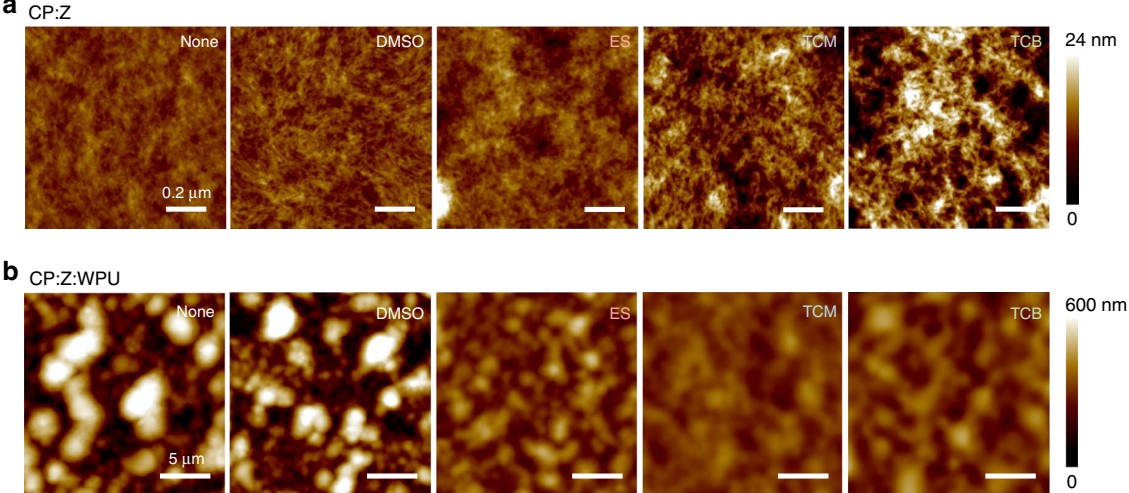

**Fig. 5 Morphological changes induced by the addition of IL.** AFM height images of **a** CP and **b** CP-WPU composite films, both with and without a different type of ILs or DMSO (Z). The CP films with ILs were washed with water before the measurements to remove excess IL on the surface. The images of CP-WPU composite films were measured on as-prepared samples, without washing. Scale bars, **a** 0.2 μm, **b** 5 μm.

phase imaging resulted in low contrast due to the softening effect of ILs (see Supplementary Fig. 11). However, after washing the samples to remove excess IL, a structure made of short nanofibrils that form a network without grain boundaries clearly appears for TCM- and TCB-added samples. The height image of the

CP-WPU composite shows large structures of 1−4 μm in size, which are not observed in the CP films (Fig. 5b and Supplementary Fig. 12). Smaller-scale images prove that these clusters consist of submicron-size grains that are assumed to originate from WPU gel particles (see Supplementary Fig. 13).

The addition of ILs disrupts the agglomeration of the WPU grains and leads to composites with much smoother surfaces. The size of the surface clusters is the smallest for the ES-added composite, which has the highest stretchability, while the RMS roughness is the lowest for TCM-added composite, which has the highest conductivity (see Supplementary Table 2 for the comparison of average cluster size and RMS roughness of the composites).

To sum up, the role of ILs that significantly improve the electrical and mechanical properties of the CP-WPU composites can be explained as follows. In the mixed CP-WPU dispersions, the particles of the different components exist almost independently without strong interaction. When the solid film is formed from the mixture, WPU particles tend to agglomerate by themselves and form micron-size clusters, while granular domains of CP are embedded in patches. As a result, the composite film exhibits poor conductivity as well as limited stretchability due to nonoptimal percolation between CP domains, imperfect connectivity between giant WPU clusters, and rigid CP domains ($E$ of CP > 500 MPa) that act as fracture points. When the IL is introduced, the counter-ion exchange between CP and IL causes a morphological change in the CP toward the crystalline nanofibrillar structure[10], which is beneficial not only to charge transport along the CP chains but also for the formation of a percolating network within the WPU matrix. It is well-known that 1D conductive fillers form a better 3D percolation in elastomeric media than spherical ones[47]. The interaction between WPU and IL is not clear since the detailed chemical structure and functional groups of WPU are not disclosed by the supplier. Nonetheless, it is clear that IL reduces the domain size and the agglomeration of WPU particles in the composite. Moreover, as a non-volatile plasticizer, IL decreases the elastic modulus of both CP (from 550 to 31 MPa) and WPU (from 3.2 to 1.7 MPa) as well as increases the stretchability of both (see Supplementary Fig. 14). Thus, the elimination of local rigid regions by softening the hard CP phase and the complete connectivity between WPU particles of uniform size result in a much more durable and stretchable composite. In contrast, DMSO has a negligible effect on hindering agglomeration of WPU particles or softening the composite since DMSO evaporates over time and does not remain in the final film[48]. Although our attempts to obtain a clear picture of how CP is incorporated in the WPU matrix by using conductive AFM (which implies contact-mode operation) failed because the inherent stickiness of WPU that we used inhibited AFM scanning in contact mode, we speculate that the fibrils of CP induced by IL addition are rather entangled in the PU network, based on the direct correlation between the electromechanical and the mechanical properties in these composite systems.

**Intrinsically stretchable organic thermoelectric module**. The elastic CP composites described above allowed us to explore intrinsically stretchable applications beyond thin film devices. Based on the excellent electromechanical property of the composite and the inherent TE property of PEDOT, we developed the first intrinsically stretchable organic TE module by using elastic thick films (≈40 μm) of CP-IL-WPU composite as $p$-type TE legs. TE modules are composed of TE legs connected electrically in series and thermally in parallel. To maximize the output power ($P_{out}$) density of TE generators, a vertical module composed of many $p$- and $n$-type legs constitutes the optimal device configuration (see Supplementary Fig. 15 for different TE device configurations)[5,21]. However, due to the absence of stretchable and high-performance $n$-type organic TE materials, we developed an intrinsically stretchable TE module composed of 10 stretchable $p$-type legs in a lateral configuration as shown in Fig. 6a, b. The use

of thick TE legs is critical for the performance of lateral TE devices since $P_{out}$ is proportional to the thickness ($\delta$) of the TE elements by the relations: $P_{out} \propto 1/R_{in}$ and $R_{in} \propto 1/\delta$, where $R_{in}$ is the internal resistance of the TE device[21].

The TE module was fabricated by laser cutting the drop-cast composite film into 10 legs of uniform thickness and transferring them onto a WPU-coated polydimethylsiloxane (PDMS) substrate. The legs were connected in series by a pattern of gold-coated titanium dioxide nanowires (Au-TiO$_2$ NWs) embedded in a PDMS substrate. The Au-TiO$_2$ NW composite was chosen as the interconnect material due to its chemical stability, low sheet resistance (<1 Ω sq$^{-1}$), good electromechanical characteristics, and negligible $S$ (<1 μV K$^{-1}$)[49]. The processing schematics and the dimensions of the module are shown in Supplementary Figs. 16 and 17. Under a temperature difference ($\Delta T$) of up to 30 K, applied to the two edges of the $p$-type 10-legs module, the generated open-circuit voltage ($V_{OC}$) is linearly proportional to $\Delta T$, resulting in a $V_{OC}/\Delta T \approx 212$ μV K$^{-1}$, which is close to 10 times the Seebeck coefficient of $S \approx 22$ μV K$^{-1}$ measured for a single TE leg (Fig. 6c). When the module is connected to varying load resistors ($R_{load}$) under $\Delta T = 10$–30 K, the $P_{out}$ reaches its maximum for $R_{load} \approx 430$ Ω, which corresponds to the total $R_{in}$ of the module (Fig. 6d). The maximum $P_{out}$ under $\Delta T = 30$ K is 25 nW, which is close to the predicted value, $P_{out\text{-}max} = V_{OC}^2/4R_{in} \approx 24$ nW.

The tensile loading up to 40% in both parallel and perpendicular directions does not affect the $V_{OC}$ of the stretchable module (Fig. 6f, i). However, the $R_{in}$ increases gradually with increasing strain due to the strain-dependent resistances of the composite legs and the interconnects (see Supplementary Fig. 18). The analysis of the strain-dependent resistances with equivalent circuits proves that the $R_{in}$ increases arose mainly from increased resistance of the Au-TiO$_2$ NW interconnects for both stretching directions, while the $R_{in}$ increase was much less pronounced for perpendicular strain (Fig. 6f, i and Supplementary Fig. 19). According to the relation $P_{out} \propto 1/R_{in}$, the $P_{out\text{-}max}$ gradually decreases with increasing strain, with 48 and 83% of the initial $P_{out\text{-}max}$ maintained for 40% parallel and perpendicular strain, respectively (Fig. 6g, j). After 7 months of storage in ambient conditions, the TE module retained the same $R_{in}$ and 86% of the original $V_{OC}$ (see Supplementary Fig. 1b) owing to the excellent stability of the CP-TCM-WPU composite (see Supplementary Fig. 20) and the Au-TiO$_2$ NWs[49]. However, thermal annealing at temperature above 100 °C led to a severe deterioration in the mechanical properties of the composite, possibly by induced phase separation and the migration of ILs, indicating that the TE module is not suitable for high temperature operation.

## Discussion

We have introduced an efficient method to formulate aqueous composite inks of PEDOT:PSS, WPU and IL, that produces elastic conductors with high conductivity (>140 S cm$^{-1}$), high stretchability (>600%), and low Young's modulus (<7 MPa). The combination of those three properties in a single material is unique and demonstrates a strategy to design an intriguing class of nanocomposites. The outstanding performance was achieved due to the addition of IL, which acts as a percolation inducer by restructuring the PEDOT complexes, impedes the agglomeration of WPU particles, and plasticizes both the PEDOT and WPU phases. The presented strategy opens up a general route for creating elastic organic conductors through the exploration of many combinations of different conducting polymers/ionic liquids/elastomers. Moreover, as exemplified in the successful demonstration of the first intrinsically stretchable organic thermoelectric module, the combination of the mechanical properties

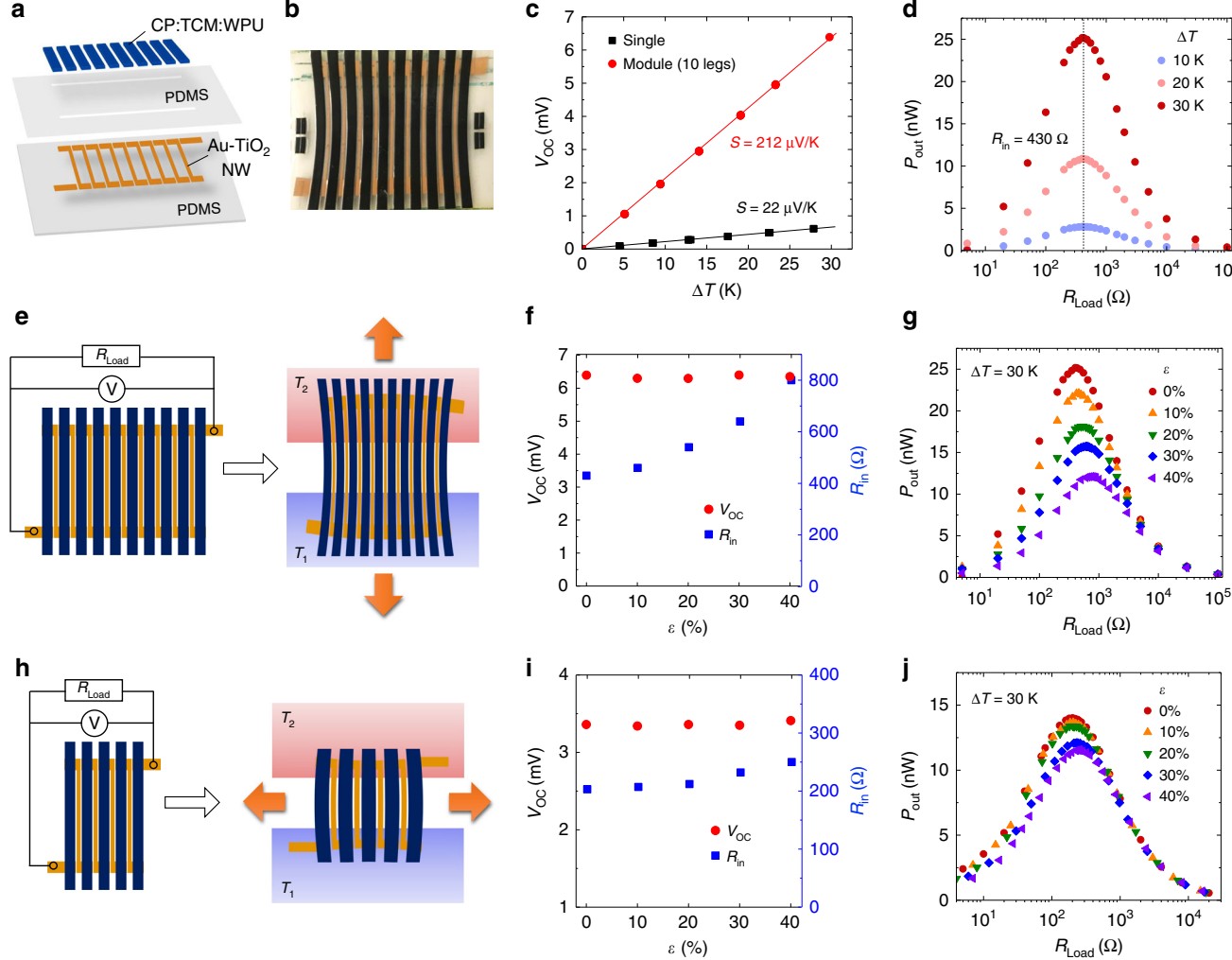

**Fig. 6 Intrinsically stretchable organic thermoelectric module. a** An architecture of the stretchable thermoelectric (TE) module. One end of each TE leg (CP:TCM:WPU = 15:25:85, w/w) is electrically interconnected with the other end of a neighboring leg through a z-shape stretchable interconnect (Au-TiO₂ nanowires). **b** Photograph of a stretchable TE module under 30% tensile strain. **c** Open-circuit voltage ($V_{OC}$) of a single leg TE device and a 10-leg TE module as a function of temperature difference ($\Delta T$). **d** Output power ($P_{out} = V_{out}^2/R_{load}$) of a TE module as a function of load resistance ($R_{load}$) for $\Delta T = 10-30$ K. The dashed line indicates an $R_{load}$ value that gives the maximum $P_{out}$, and thereby corresponds to the total internal resistance ($R_{in}$) of a module. **e–g** TE module stretched parallel with the thermodiffusion. $T_1 = 23$ °C, $T_2 = 53$ °C. **f** $V_{OC}$ and $R_{in}$ vs. tensile strain ($\varepsilon$) applied to a module for $\Delta T = 30$ K. **g** $P_{out}$ vs. $R_{load}$ under different strains for $\Delta T = 30$ K. **h–j** TE module stretched perpendicular to the thermodiffusion and corresponding $V_{OC}$ and $R_{in}$ vs. $\varepsilon$ and $P_{out}$ vs. $R_{load}$ under different strains for $\Delta T = 30$ K.

of the developed PEDOT composites and the potential of PEDOT for various applications (electronic, electrochemical, ionic, optical, and TE devices)[50] are expected to dramatically expand PEDOT's applicability to mechanically demanding applications, including on or inside human body, textiles, and any kind of irregular or moving surfaces.

## Methods

**Materials**. An aqueous dispersion of PEDOT:PSS (Clevios™ PH1000; solid content ≈ 1.1−1.3 wt%; weight ratio of PEDOT to PSS≈1:2.5; pH~2) was purchased from Heraeus. A WPU dispersion (ALBERDINGK® U4101; solid content ≈ 39−41 wt%; elongation at break~1400%; pH ≈ 7.0−8.5) was obtained from Alberdingk Boley. ALBERDINGK® U4101 is an aqueous, anionic dispersion of an aliphatic polyether-polyurethane without free isocynate groups. 1-Ethyl-3-methylimidazolium coordinated with ethyl sulfate (EMIM ES) and tricyano-methanide (EMIM TCM) were purchased from Sigma-Aldrich and Tokyo Chemical Industry, respectively. 1-Ethyl-3-methylimidazolium tetracyanoborate (EMIM TCB) was synthesized as described in the previous literature[10]. DMSO was purchased from Sigma-Aldrich. An ammonia hydroxide solution 28% was purchased from BASF. All chemicals were used as received without further purification.

**Preparation of composite dispersions and films**. All ILs were diluted to 1.48 wt% with deionized (DI) water before being added into PEDOT:PSS dispersions. A solution of a different type of ILs with varying amounts was added dropwise to the PEDOT:PSS dispersion. During blending, the PEDOT:PSS dispersion was stirred at 400 rpm and continued to be stirred for >12 h. For the dispersions containing DMSO, 5.6 wt% of DMSO was added to the PEDOT:PSS dispersion and then stirred at 400 rpm for >1 h. Before adding a WPU dispersion, 0.15 wt% of an ammonia solution was added with respect to the PEDOT:PSS dispersion. Varying amounts of a WPU dispersion were added to the different dispersions (PEDOT:PSS, PEDOT:PSS-IL, PEDOT:PSS-DMSO) prepared as mentioned above while the dispersions were stirred at 400 rpm, and then the entire mixtures were stirred for >2 h. The dispersions were spin-cast or drop-cast onto cleaned glass or silicon substrates and then dried in a vacuum desiccator at room temperature for the characterization of the films with different compositions.

**Mechanical, electrical, thermoelectric, and electromechanical characterization of composite dispersions and films**. For mechanical and electromechanical analysis, free-standing films (width ≈ 2 mm; length ≈ 10−20 mm, thickness ≈ 30 −60 μm) were clamped on a motorized linear stage (X-LSQ300A-E01, Zaber) either with gold-coated 4-point contact pads for resistance vs. strain measurements or with a force gauge (M5-2, Mark-10) for stress vs. strain measurements. The strain rate was 1% s⁻¹ for the resistance-strain measurements and the maximum strain tests, and 2% s⁻¹ for the cycling tests. Resistances under strain were monitored by using a

Keithley 2701 Ethernet Multimeter data acquisition system. For dc conductivity measurements, sheet resistances were measured by Van der Pauw method on films spin-coated on glass substrates (area ≈ 20 × 20 mm², thickness ≈ 0.1–0.5 μm). A current was applied while the voltage was measured in a 4-wire sensing configuration using a Keithley 2400 Source Meter unit. The film thickness was determined as the average value taken from four positions by using a surface profilometer (Dektak 3ST, Veeco). Seebeck coefficient measurements were carried out using a fully automated LabVIEW controlled setup where the sample was placed on two Peltier elements with adjustable separation. One of the Peltier elements was heated up to 53 °C, while the other one was kept at room temperature (23 °C). The thermoelectric voltage generated between Au-TiO$_2$ NW contact pads placed at two edges of the sample was recorded using a Keithley 2182A nanovoltmeter, while the temperature was measured using thermocouples. For all samples, $V_{OC}/\Delta T$ shows a linear slope in the measured temperature range (23−53 °C). Note that we measured S for all samples 3 or 4 days after the sample preparation due to the time dependence of S with the recovery of oxidation level over time (see Supplementary Fig. 1b). Changes in the dimension (width and thickness) of a free-standing film under different strains were determined by analyzing optical images obtained using an optical 3D surface profiler (PLu neox, Sensofar).

**Structural characterization of composite dispersions and films**. An ALV-CGS-5022F goniometer system (ALV GmbH, Langen) with a 22 mW 633 nm HeNe laser was used for DLS. To prepare samples for DLS measurements, dispersions without WPU were diluted in DI water to make the solid content of PEDOT:PSS to be 0.0016 wt%. Dispersions with WPU were diluted in DI water to make the solid content of WPU to be 0.001 wt%. Samples were contained in 10 mm diameter cylindrical cuvettes, immersed in a refractive index-matching toluene bath. The toluene was temperature-controlled to 22 ± 0.02 °C using a water circulator. Samples were equilibrated in the circulating water for at least 10 min before measurements. The system was operated in pseudo-cross-correlation mode, with scattered light collected into a single-mode optical fiber, split to two avalanche photodiodes (Perkin-Elmer). A dual multiple-tau correlator with 328 channels (ALV-6010-160) yields the time-correlation function of the scattered intensity. A constrained regularization method (CONTIN, supplied with the correlator software) was used to calculate correlation time distributions, which were re-calculated to size distributions via the Stokes-Einstein relationship[51]. XRD $\theta$–2$\theta$ scans were carried out in a PANalytical X'Pert PRO diffractometer system equipped with a Cu $K_\alpha$ source operated at 45 kV and 40 mA. The incident optics was a Bragg–Brentano module including a 0.5 divergence slit and a 0.5 anti-scatter slit, and the diffracted optics included a 5.0-mm anti-scatter slit and 0.04-rad soller slits. The PreFIX detector was set to one-dimensional scanning line mode. Step sizes and collection times per step were 0.05 and 2 s, respectively. The AFM data were recorded in ambient conditions either using a Dimension 3100 microscope equipped with a Veeco Di Nanoscope IIIa controller (for CP:Z samples) or using a Bruker Dimension ICON microscope equipeed with a Controller V (for CP:Z:WPU samples) from Bruker Nano. Inc. (Santa Barbara, CA). For AFM tapping mode, silicon probes with a resonance frequency of 300 kHz and a spring constant of 40 N m$^{-1}$ were used.

**Synthesis of Au-TiO$_2$ NWs**. TiO$_2$ nanowires (Novarials, NovaWire-TiO-10-RD, length=10 μm, 0.72 mg), hydroxylamine (Sigma-Aldrich, 50%, 120 μL), and poly (vinylpyrrolidone) (Sigma-Aldrich, MW 55 kDa, 1.32 g) were mixed with DI water to a final volume of 40 mL. Gold(III) chloride solution (Sigma-Aldrich, Au 17 wt%, 244 μL) was diluted in DI water to a final volume of 40 mL. The gold solution was added with a syringe pump into the TiO$_2$ NW solution under stirring (4 mL at 5.6 mL min$^{-1}$, then 36 mL at 2.8 mL min$^{-1}$). Hydrochloric acid (Sigma-Aldrich, 37%, 4 mL) was added 5 min after completion and the dispersion was matured at 80 °C for 15 min until its appearance changed to light brown. The dispersion was put through a 40 μm mesh cell strainer nylon sieve to remove large aggregates. The dispersion was stored for at least 1 day before use.

**Fabrication and characterization of intrinsically stretchable thermoelectric modules**. A pattern of Au-TiO$_2$ NWs with nine interconnects in the middle and two contact pads at the two edges was prepared for the fabrication of a 10-legs TE module by wax-assisted vacuum filtration (8.8 mL Au-TiO$_2$ NW solution, open filter area ≈ 213 mm², sheet resistance ≈ 0.28 Ω sq$^{-1}$)[52]. The dimension of a pattern is described in Supplementary Fig. 17. A pattern with only single interconnect and another pattern with only two contact pads were also prepared for the characterization of the electromechanical properties of Au-TiO$_2$ NWs and the fabrication of a 1-leg TE device, respectively. A PDMS mixture (Sylgard 184, Dow Corning, 1:10) was spin-cast (30 rps, 30 s) on a cured PDMS substrate (area ≈ 50 mm × 60 mm, thickness ≈ 0.2 mm) supported on a glass slide and then semi-cured (70 °C, 14 min). The Au-TiO$_2$ NW pattern prepared on a filter membrane was placed on a semi-cured PDMS sample and transferred by hot-pressing (70 °C, 20 min). A 25 μm poly(ethylene naphthalate) (PEN) foil was used to mask the contact areas and a layer of PDMS was spin-coated on top (60 rps, 30 s). The foil was removed and the sample was cured (80 °C, >2 h). The sample was treated with UV/O$_3$ for 30 min and then spin-coated with a thin layer of WPU (10 wt% diluted U4101, 3000 rpm, 1 min). This WPU layer improved the mechanical adhesion

between interconnects/substrate and composite TE legs while still keeping the contact areas of Au-TiO$_2$ NWs open in the air. A composite film drop-cast on a glass slide (1 mL on 25 mm × 40 mm area) was cut into 10 legs (for each leg, width ≈ 1.8 mm, length ≈ 28 mm) using a laser cutter (Trotec Speedy 300 flexx) equipped with a pulsed fiber laser and then transferred to the WPU-coated Au-TiO$_2$ NW-embedded PDMS sample. The finished TE module was annealed (60 °C, >20 min) to improve contact between Au-TiO$_2$ NWs and composite TE legs before measurements. The processing schematics were also drawn in Supplementary Fig. 16. The two edges of a stretchable TE module were fixed on glass slides by Kapton tape with an initial distance of 25 mm. The module was stretched by placing another glass slides with different distances between two fixing slides. The module was torn when strain increased over 50% due to the limited stretchability of PDMS at a low aspect ratio of 25:60 (length:width). A 5-leg TE module was fabricated via the same process but with an additional encapsulation layer of a WPU-coated PDMS substrate (area ≈ 50 mm × 60 mm, thickness ≈ 0.2 mm) laminated on top of the module. The number of legs was reduced due to the limited size of the Peltier elements to cover the module under perpendicular stretching, while the top encapsulation layer was introduced to promote the TE legs to conform with PDMS substrates. A 5-leg TE module was stretched in the similar manner in perpendicular direction with an initial distance of 36 mm. Thermovoltage measurements were carried out in the same way for S measurements. For thermopower measurements, a resistance decade box (Type-BE 11, Betatron) was connected in parallel and the power output was calculated from $P_{out} = V_{out}^2/R_{load}$.

All experiments were conducted in the clean room at steady temperature (≈21 °C) and humidity (≈47% RH) conditions.

## Data availability
The data that support the findings of this study are available from the corresponding authors upon reasonable request.

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

## Acknowledgements

This work was financially supported by the Knut and Alice Wallenberg Foundation (Tail of the sun), the Göran Gustafsson Foundation, the Swedish Foundation for Strategic Research and the Swedish Government Strategic Research Area in Materials Science on Advanced Functional Materials at Linköping University (Faculty Grant SFO-Mat-LiU No. 2009-00971). Research in Mons is supported by the Belgian National Science Foundation (FNRS, PDR grant T.1004.14 - ECOSTOFLEX). Ph.L. is FNRS senior research associate. We thank Thai Cuong Nguyen for discussion and trials. Open access funding provided by Linköping University.

## Author contributions

N.K., X.C., and K.T. conceived the idea. N.K. formulated the composites and conducted the electrical, mechanical, electromechanical characterization. S.L. synthesized and patterned Au-TiO$_2$ NWs. N.K. and S.L. fabricated the TE modules. N.K. and I.P. conducted the thermoelectric characterization. D.A.M. contributed to formulate the composite. S.K. synthesized EMIM TCB. T.E. conducted the DLS measurements. N.K. conducted the XRD measurements. N.K. and V.G. conducted the AFM measurements. Ph.L. and R.L. helped with the AFM measurements. N.K., X.C. and K.T. wrote the paper. X.C. and K.T. supervised the work. All authors contributed to the finalization of the paper.

## Competing interests

The authors declare no competing interests.
