## [Peer Review File · Nature Communications]

Reviewers' comments:

Reviewer #1 (Remarks to the Author):

The authors report the fabrication of a novel PEDOT-ionic liquid composite that exhibits high conductivity, elasticity, stretchability, and a low young's modulus. The authors attribute the high performance of their material to favorable nano- and micro- scale percolation networks, and plasticizing effect of the ionic liquid, and then use this material to fabricate an intrinsically stretchable organic thermoelectric module. To support their hypothesis of the importance of percolation networks, the authors investigated the properties of their composite over a wide range of composition, and measured the size distribution of aggregates in the aqueous dispersions of the composite that they used to fabricate films of their materials, as well as performed atomic force microscopy measurements to visualize the surface morphology of their films.

Although intrinsically stretchable thermoelectric materials have been previously demonstrated, the fabrication of a thermoelectric module with tens of micron thick legs from such a material, along with the increased electrical conductivity and supported hypothesis for this increase, represents a significant advance for the understanding and utility of these materials. This manuscript is both sufficiently novel, and relevant to a broad audience to justify publication in Nature Communications after addressing the following comments:

1)

Although the authors report electrical properties as a function of strain and reference cycling tests for the physical properties of their materials, it is unclear how the electrical properties evolve with strain cycling. If the conducting polymer is formed of a percolated network of nanofibrils, it seems reasonable that cycling would induce changes in this morphology that would affect the electrical conductivity or Seebeck coefficient. Additionally, although electrical conductivity can be inferred from the change in resistance and geometric changes for strain up to 100%, the Seebeck coefficient is shown only up to strains of 40%. If the authors have additional data on electrical properties as a function of cycle, and Seebeck coefficients for larger strain values, these data would be useful for a reader to understand the importance of the demonstrated thermoelectric module.

2)

Similar to above, although the resistance has increased and therefore current output of the thermoelectric module decreased by a factor of approximately two for tensile strains of 40%, these changes are explained by geometric changes in the material only if it behaves as a material whose resistivity is an intensive material property. Given that percolation through a network of fibers is a more complex model of conduction, the authors should comment on the validity of their assumptions when using an effective medium model to determine the conductivity of the embedded conducting polymer phase, or uniform geometric model that invokes intensive material properties, such as for 2(h) and S3.

3)

The DC conductivities were measured on spun-cast films that were significantly thinner than the legs of the drop-cast thermoelectric modules. The authors should comment on potential differences in morphologies of materials deposited in these ways, and specifically how those differences might affect a percolation model of conduction.

Reviewer #2 (Remarks to the Author):

This is an interesting paper on the development of flexible thermoelectric based on PEDOT. Inclusion of ionic liquids (ILs) into the material make possible the development of stretchable

thermoelectrics. The work has been carried out very well and the results should be of interest to the thermoelectric community. Unfortunately, while I think the work is good, I don't believe it has the level of impact and novelty that one would expect for the journal. I don't believe it will impact on thinking in the field to a sufficient level.

On a scientific level, I note that the authors haven't really addressed loss or leaching of the plasticiser from the materials. In the development of composite materials based on ILs often must determine the stability, dispersion and loss of the material in/from the host material. I think this is an important issue that must be addressed.

Reviewer #3 (Remarks to the Author):

The authors report on the stretchable PEDOT:PSS-IL-WPU composite for the stretchable thermoelectric (TE) module. In this manuscript, the solution processable TE ink is prepared using the PEDOT:PSS-IL-WPU composite and this composite exhibits quite good electrical and mechanical properties. However, materials based on PEDOT:PSS with elastomer have widely been studied even with their use in stretchable electronics. Furthermore, the addition of IL for the TE application of PEDOT:PSS is a routine process. The fabricated TE module in parallel direction without n-type materials also shows quite low output power despite the good mechanical stability. Although the paper is well written and presented in a scholarly manner with few exceptions, I believe this manuscript, while interesting, is insufficient to provide readers with useful information in the field of thermoelectronics. Therefore, I strongly recommend rejection for publication in Nature Communications.

- 1) What is the change in the Seebeck coefficients depending on the composition ratio of three substances, PEDOT:PSS, IL, and WPU?
- 2) On line 206, the author referred to a paper that is not yet published.
- 3) On line 209, the author claims that the conductivity of composite decreased by dedoping is recovered via chemical equilibrium with the oxygen in air. However, no evidence or reference can be found to support the author's claim in the manuscript. Provide the appropriate evidences or refer relevant papers.
- 4) On line 318, are the Au-TiO₂ NWs stretchable? (If they are stretchable enough, please provide the proper evidence.) Resistance of the Au-TiO₂ NWs is significantly increased with an increase of tensile strain in Figure S15. I wonder the drop in the output power of the TE module in Figure 5f is closely related to increased resistance of the Au-TiO₂ NWs.
- 5) In page 30, the authors evaluated the TE properties of the stretchable TE module in only one direction. Since the active part of the TE module shown in Figure 5 has a long rectangular shape with anisotropy (not a square), they should be characterized in both perpendicular and parallel direction to elongation.
- 6) On line 650, the elasticity of TE module with the composite is different from that of the pristine active material because the TE module includes both substrate and electrode. Is there no effect of the substrate and the electrode on the performance of the active layer? Also, direction of stretching should be indicated with an arrow in Figure 5 for the reader's understanding.
- 7) On line 305, for wearable and stretchable TE modules, vertical configuration is more suitable than parallel one due to its easy build-up of the temperature gradient and low electrical resistance. If the PEDOT:PSS-IL-WPU composite forms the enough thickness, vertical TE module can be prepared by using only p-type TE legs. Is the thickness of the PEDOT:PSS-IL-WPU films (~40 μm) sufficient for the TE legs of vertical configuration? If it is impossible, how can the thickness be increased or controlled? How can the electrical conductivity and mechanical properties be optimized depending on thickness?

Summary of Revisions Made in Responses to Reviewers' Comments

=====

Responses to Reviewer #1's comments

=====

[Comment #1] The authors report the fabrication of a novel PEDOT-ionic liquid composite that exhibits high conductivity, elasticity, stretchability, and a low young's modulus. The authors attribute the high performance of their material to favorable nano- and micro- scale percolation networks, and plasticizing effect of the ionic liquid, and then use this material to fabricate an intrinsically stretchable organic thermoelectric module. To support their hypothesis of the importance of percolation networks, the authors investigated the properties of their composite over a wide range of composition, and measured the size distribution of aggregates in the aqueous dispersions of the composite that they used to fabricate films of their materials, as well as performed atomic force microscopy measurements to visualize the surface morphology of their films.

Although intrinsically stretchable thermoelectric materials have been previously demonstrated, the fabrication of a thermoelectric module with tens of micron thick legs from such a material, along with the increased electrical conductivity and supported hypothesis for this increase, represents a significant advance for the understanding and utility of these materials. This manuscript is both sufficiently novel, and relevant to a broad audience to justify publication in Nature Communications after addressing the following comments:

[Response #1] We thank Reviewer #1 for evaluating our paper and appreciating the novelty and significance of our work on developing highly conductive elastic conducting polymer composites and demonstrating the first intrinsically stretchable organic thermoelectric modules based on the composites. As Reviewer #1 has pointed out, intrinsically stretchable thermoelectric materials based on PEDOT:PSS/elastomer composites and PEDOT:PSS/plasticizer composites, which we have also discussed in detail in the introduction section, have been previously reported. However, none of those materials could constitute the real module applications due to low conductivity of elastomeric composites or poor mechanical property of plasticizer composites as well as the lack

of suitable stretchable interconnects. So, we agree with Reviewer #1 that our work represents a significant advance in the field of stretchable thermoelectrics. We tried our best to address all the valuable comments from Reviewer #1.

[Comment #2] Although the authors report electrical properties as a function of strain and reference cycling tests for the physical properties of their materials, it is unclear how the electrical properties evolve with strain cycling. If the conducting polymer is formed of a percolated network of nanofibrils, it seems reasonable that cycling would induce changes in this morphology that would affect the electrical conductivity or Seebeck coefficient. Additionally, although electrical conductivity can be inferred from the change in resistance and geometric changes for strain up to 100%, the Seebeck coefficient is shown only up to strains of 40%. If the authors have additional data on electrical properties as a function of cycle, and Seebeck coefficients for larger strain values, these data would be useful for a reader to understand the importance of the demonstrated thermoelectric module.

[Response #2] According to the Reviewer #1's comment, we examined the electrical properties of the optimized composites (CP:TCM:WPU=15:25:85, w/w) as a function of stretching cycle at three different tensile strains (20%, 40%, and 60%). As shown in Fig. R1 (Fig. 3e in the revised manuscript), the composite shows excellent cycling stability when stretched to 20% and 40% 1000 times. The initial increase in resistance (R) caused by the separation between CP nanofibrils with geometric changes declined gradually over cycles presumably due to the alignment and reconnection of nanofibrils (*Sci. Adv.* 2017; 3 : e1602076). Similar behavior was also observed for AuNW network (*Adv. Mater.* 2018, 1706520). R of the composite became even less than the original R when released after 1000 cycles at 20% or increased by only 6% after 1000 cycles at 40%. However, further rearrangement of conductive fibrils under 60% strain caused plastic deformation in order to accompany the stress focused on the weak points of the composite (e.g. relatively thinner parts) and the repeated stress resulted in the complete tear of these points after approximately 200 cycles (see Fig. R2). Nonetheless, when the composite was supported on a PDMS substrate as in the case for device applications, the stress is applied more uniformly across the substrate and the process for plastic deformation is much slowed down. As a result, the supported composite could withstand 1000 stretching cycles at 60% strain with the increase in R

by 70%.

Figure R1. Resistance changes (R/R_0) of the optimized composite (CP:TCM:WPU=15:25:85) upon cyclic tensile stretching (blue lines). R/R_0 of the optimized composite supported on a PDMS substrate under cyclic strain at 60% is also shown (gray line).

Figure R2. a) Resistance changes (R/R_0) of the free-standing optimized composite (CP:TCM:WPU=15:25:85) under cyclic strain at 60%. b) Optical microscope images of the torn parts of the composite after 200 cycles.

Seebeck coefficients of the optimized composite for larger strain values up to 100% were also examined as shown in Fig. R3 (Fig. 3c in the revised manuscript). Seebeck coefficients were retained nearly the same under strains up to 80% and slightly decreased at 100% strain, indicating that the density of percolation paths and the corresponding density of states are maintained in this

strain range (*Adv. Electron. Mater.* 2019, 5, 1800918 & *Nat. Mater.* 13, 190-194 (2014), and please refer our response to Reviewer #3's comment #2).

Figure R3. Seebeck coefficients (S) of the composites with a different type of ILs and S vs. tensile strain (ϵ) for the optimized composite.

In response to Reviewer #1's comment, we included the new results in the manuscript and supplementary information and revised the main text as shown below.

(on page 9) S does not vary much depending on the type of ILs, while the inverse relationship between σ_{dc} and S of the composite still holds with different ILs that cause different percolative networks. The S of the composite with TCM was nearly constant for strains up to 80% and decreased slightly at 100% strain.

(on page 9) The R/R_0 of the composite with TCM shows excellent cycling stability when stretched up to 40% (Fig. 3e). R became even less than R_0 when released after 1000 stretching cycles at 20% or increased by only 6% after 1000 cycles at 40%. The decline in R over cycles is ascribed to the alignment of CP fibrils¹⁸. However, the viscoelastic property of the composite caused partial plastic deformation at 60% strain and repeated stresses focused on weak points resulted in the complete tear after approximately 200 cycles (see Supplementary Fig. S6).

Nonetheless, when supported on a PDMS substrate, more uniformly distributed stresses across the substrate suppress plastic deformation so that the composite could withstand 1000 stretching cycles at 60% with a R/R_0 of only 1.7.

[Comment #3] Similar to above, although the resistance has increased and therefore current output of the thermoelectric module decreased by a factor of approximately two for tensile strains of 40%, these changes are explained by geometric changes in the material only if it behaves as a material whose resistivity is an intensive material property. Given that percolation through a network of fibers is a more complex model of conduction, the authors should comment on the validity of their assumptions when using an effective medium model to determine the conductivity of the embedded conducting polymer phase, or uniform geometric model that invokes intensive material properties, such as for 2(h) and S3.

[Response #3] The reviewer is correct that the complex geometry of the CP phase within the composite is unknown. However, by considering the orientation factor (conduction pathways are not aligned in the macroscopic direction of conduction) from an ideally connected geometry, e.g. the fiber contact model, we find that the effective conductivity of the CP phase is approximately equal to that of pure CP thin films. This would not be the case if the CP phase constituted a poorly percolating network, as in that case a lot of the CP material would not contribute to the conduction. Thus, the analysis indicates that our CP network exhibits close to ideal percolation behavior. We have clarified the assumptions for this analysis in the text.

(on page 10) Due to the orientation of the conducting segments within a composite, the σ_{dc} is typically lower in a composite than in a pure conductor. (*Int. J. Hydrogen Energy* 42, 9262 (2017)) Even for well-connected systems, like meshes of randomly oriented nanowires or connected tubes of 45° angle with respect to the direction of conduction (fiber contact model), the σ_{dc} is reduced by 50 %. Although the exact geometry of the CP phase in our composite is unknown, by assuming that it possesses the same orientational characteristics as the fiber geometries described above, the effective σ_{dc} for the CP phase would be approximately 2340 S cm⁻¹. This is nearly the same value as for the CP film without WPU (≈ 2350 S cm⁻¹, see Supplementary Fig. S3), indicating that most of the PEDOT network within the composite is

contributing to the conduction.

To avoid possible misleading, we showed in Fig. 3b and Fig. S5 (Fig. 2h and Fig. S3 in the previous version) the changes in R and geometry of the composites as a function of strain, which can be directly measured, rather than showing the conductivity change that can be estimated from R and geometric changes with large errors. The R_G/R_0 (*exp.*) values plotted in Fig. 3b come from simple mathematics ($R \propto \text{length}/(\text{width} \times \text{thickness})$) based on average values for real geometric changes of the composite shown in Fig. S5 without employing any geometric model with the comparison to the case for rubber having a Poisson's ratio of 0.5. To clarify this, we revised the text in the Supplementary Information as shown below.

(on page 7 in Supplementary) R_G/R_0 (*exp.*) in Fig. 3b was estimated from the average values measured for dw/w_0 and dt/t_0 plotted in Fig. S5 according to the relation:

$$\frac{R_G}{R_0} = \frac{1 + dl/l_0}{(1 + dw/w_0)(1 + dt/t_0)}$$

[Comment #4] The DC conductivities were measured on spun-cast films that were significantly thinner than the legs of the drop-cast thermoelectric modules. The authors should comment on potential differences in morphologies of materials deposited in these ways, and specifically how those differences might affect a percolation model of conduction.

[Response #4] Spin-coating is a useful technique to form thin (<1 μm) films having uniform thicknesses. Centrifugal forces that uniformly spread the dispersion across the substrate as well as fast drying ensure the formation of homogeneous films, while centrifugal forces often induces the orientation of polymer molecules that enhances the charge carrier mobility (*Journal of Nanomaterials*, Volume 2017, Article ID 3624750, 18 pages). In contrast, drop-casting takes much longer drying time with differences in drying rates across the substrate. Therefore, there could be concentration gradients or more phase separation (*ACS Appl. Mater. Interfaces* 2015, 7, 35, 19764) between different components that results in less homogeneous films with larger variations in thickness compared to spin-coated films. Accordingly, drop-cast films (>10 μm)

usually show lower conductivity values than spin-coated films of the same materials. For instance, the conductivity values reported for the most representative PEDOT:PSS formulations (i.e. PEDOT:PSS with 5 wt% of DMSO) were different from drop-cast films (≈ 660 S/cm, *Phys. Rev. Lett.* 109, 106405 (2012)) and spin-coated films (≈ 890 S/cm, *ACS Appl. Mater. Interfaces* 2016, 8, 11629) by 1.35 fold. Our composite (CP:TCM:WPU=15:25:85) also shows lower conductivity for the drop-cast films (≈ 100 S/cm) used as thermoelectric legs than the spin-coated films (≈ 140 S/cm,) by 1.4 fold. Based on the similar difference in conductivities observed for the representative PEDOT:PSS formulations, we assume that the effect of the different deposition methods is related to morphological differences induced by the film formation process. Note however that the conductivity stays in the same order of magnitude, which is a clear indication that the microscopic morphology of the CP governing the charge transport in the insulating matrix is not completely different between spin-coated and drop-cast films. To clarify this, we revised our manuscript as shown below.

(on page 10) Compared with the spin-coated film ($\sigma_{dc} \approx 140$ S cm⁻¹), the drop-cast film of the optimized composite has only a slightly lower σ_{dc} (≈ 100 S cm⁻¹), a similar reduction has been observed for CP-DMSO films^{41,42}; which indicates that thicker free-standing drop-cast films used below in thermoelectric generators have likely similar transport mechanism and microscopic morphology as spin-coated films.

=====
Responses to Reviewer #2's comments
=====

[**Comment #1**] This is an interesting paper on the development of flexible thermoelectric based on PEDOT. Inclusion of ionic liquids (ILs) into the material make possible the development of stretchable thermoelectrics. The work has been carried out very well and the results should be of interest to the thermoelectric community. Unfortunately, while I think the work is good, I don't believe it has the level of impact and novelty that one would expect for the journal. I don't believe it will impact on thinking in the field to a sufficient level.

[**Response #1**] We thank Reviewer #2 for evaluating our paper and appreciating the excellence of our work. Along with the recent paradigm shift from flexible electronics toward stretchable electronics, an appealing challenge is to realize intrinsically stretchable thermoelectric modules that can be attached conformably to our body to power wearable electronics using body heat. Despite increasing efforts to develop intrinsically stretchable thermoelectric materials based on organic materials focusing on PEDOT, the failure to simultaneously achieve both good thermoelectric properties (electrical conductivity, Seebeck coefficient) and excellent mechanical properties (stretchability, elasticity, softness) in the form thick enough to be used in thermoelectric modules has been limiting the demonstration of intrinsically stretchable organic thermoelectric modules. Instead, stretchable thermoelectric generators have been recently demonstrated based on deformable coil structures or tearing processes of rigid inorganic materials with the limitations such as out-of-plane deformation or severe degradation of output power upon stretching (*Sci. Adv.* 2018, 4 : eaau5849; *Energy Environ. Sci.* 2016, 9, 1696). In our work, for the first time, we demonstrated truly intrinsically stretchable organic thermoelectric modules by developing a novel composite material that simultaneously possesses high electrical conductivity, high stretchability, and elasticity with low Young's modulus in the thick free-standing form, properties which never have been achieved simultaneously in any organic conductor previously. Our novel composite material enables not only thermoelectric generators but also other PEDOT-based devices requiring thick films (supercapacitors, actuators, biosensors, etc.) to be used in mechanically demanding conditions. Moreover, the presented strategy to successfully combine a number of properties in a single composite material and the implication

revealed by characterizing the first demonstrated intrinsically stretchable organic thermoelectric modules (please refer our response to Reviewer #3's comment #1 for details) would have significant impacts to design a novel class of nanocomposites and for the further development of intrinsically stretchable thermoelectric modules. Therefore, we believe that our work will be of interest to the broad readership in the fields of material/polymer science, organic electronics and thermoelectrics.

[Comment #2] On a scientific level, I note that the authors haven't really addressed loss or leaching of the plasticiser from the materials. In the development of composite materials based on ILs often must determine the stability, dispersion and loss of the material in/from the host material. I think this is an important issue that must be addressed.

[Response #2] We appreciate the Reviewer #2's valuable comment. The stability of the material upon stretching cycles is discussed in the response #2 of Reviewer #1. Here, we focus on the stability of the material with time and temperature. As the reviewer concerned, loss of plasticizers is an important issue that can deteriorate the physical properties of composites. Although the negligible volatility of ionic liquids compared with traditional plasticizers (*European Polymer Journal* 39 (2003) 1947) solves this issue of loss of plasticizers through evaporation, migration of plasticizers inside or out of the host material still can lead to loss of flexibility caused by the reduced dispersion or leaching of plasticizers. To assess the stability of our composite, we examined optical microscope images and stress-strain curves of the composites (CP:TCM:WPU=15:25:85) as a function of storage time and annealing temperature. As shown in Fig. R4, there is no notable change in the surface morphology over time up to 7 months except the creation of tiny lumps on the surface. Meanwhile, the 7-months-old composite retains its initial conductance (e.g. R of the sample measured 7 months after the sample preparation is nearly the same as that measured 3 days after the sample preparation) and possesses even higher elongation at break than the fresh sample. Therefore, we assume that tiny lumps may originate from small-molecular polyurethanes that migrated from the bulk to the surface over time, while this migration would not have the effect on the electrical percolation or the mechanical connection between large-molecular polyurethanes. The composite shows also excellent stability against thermal annealing at 80 °C as proven by the similar morphology and mechanical property

to the sample before thermal annealing. However, thermal annealing at 100 °C led to significantly different morphology in optical microscope images in conjunction with a severe deterioration in its mechanical property (i.e. elongation at break decreased down to 27%). Based on haze in the image and the poor mechanical property observed for the composite without ionic liquids, we assume that the phase separation between different components and the migration of ionic liquids to the surface may occur at this temperature, implying that thermoelectric modules made of our composites are not suitable for high temperature operation above 100 °C. To address this issue clearly, we revised the manuscript and supplementary information as shown below.

(On page 15) After 7 months of storage in ambient conditions, the TE module retained the same R_{in} and 86% of the original V_{OC} (see Supplementary Fig. S1b) owing to the excellent stability of the CP-TCM-WPU composite (see Supplementary Fig. S20) and the Au-TiO₂ NWs⁴⁹. However, thermal annealing at temperature above 100 °C led to a severe deterioration in the mechanical properties of the composite, possibly by induced phase separation and the migration of ILs, indicating that the TE module is not suitable for high temperature operation.

(on page 20 in Supplementary) The similar surface morphology and mechanical property preserved after 7 months storage time or thermal annealing at 80 °C show the excellent stability of the composite with a stable dispersion of ionic liquid plasticizers. Considering the preserved electrical and mechanical properties over time, the tiny lumps observed on the surface of old samples may result from the migration of small-molecular polyurethanes. Thermal annealing at 100 °C led to a significantly different morphology in conjunction with a severe deterioration in mechanical property (i.e. elongation at break decreased down to 27%). Based on haze in the image and poor mechanical property observed for the composite without ionic liquids, we assume that the phase separation between different components and the migration of ionic liquids to the surface may occur at this temperature.

Figure R4. Optical microscope images and stress (σ)-strain (ϵ) curves of the optimized composite (CP:TCM:WPU=15:25:85, w/w) films used for TE legs after storage time up to 7 months in ambient conditions or after heating up to 100 °C.

=====

Responses to Reviewer #3's comments

=====

[**Comment #1**] The authors report on the stretchable PEDOT:PSS-IL-WPU composite for the stretchable thermoelectric (TE) module. In this manuscript, the solution processable TE ink is prepared using the PEDOT:PSS-IL-WPU composite and this composite exhibits quite good electrical and mechanical properties. However, materials based on PEDOT:PSS with elastomer have widely been studied even with their use in stretchable electronics. Furthermore, the addition of IL for the TE application of PEDOT:PSS is a routine process. The fabricated TE module in parallel direction without n-type materials also shows quite low output power despite the good mechanical stability.

Although the paper is well written and presented in a scholarly manner with few exceptions, I believe this manuscript, while interesting, is insufficient to provide readers with useful information in the field of thermoelectronics. Therefore, I strongly recommend rejection for publication in Nature Communications.

[**Response #1**] We thank Reviewer #3 for evaluating our paper and constructive comments that were helpful to greatly improve the scientific level and impact of our paper. As the reviewer pointed out and also as we have already discussed in the introduction of our manuscript, the composites of PEDOT:PSS with elastomers have been studied for stretchable electronics due to the decent stretchability and elasticity, but without successfully simultaneously achieving high electrical conductivity (few S/cm). On the other hand, ILs act not only as secondary dopants to significantly enhance the conductivity of PEDOT:PSS to reach the order of 1000 S/cm but also as plasticizers to improve the stretchability of PEDOT:PSS thin film carried by an elastomer substrate. Note that drop-cast thick films of PEDOT:PSS-IL composites are plastic, and not elastic, and fractured under strain below 25% (*ACS Appl. Mater. Interfaces* 2017, 9, 819; *Chem. Mater.* 2019, 31, 3519). Hence, up to now nobody has succeeded to create a highly conducting free-standing thick polymer layer that displays elasticity (and not only stretchability). That is the reason why nobody could yet demonstrate intrinsically stretchable TE modules to produce meaningful output power under mechanical stretching and this is the novelty of our manuscript. Although the output power can be significantly enhanced by altering module architectures from

lateral to vertical configurations as discussed in our response to Reviewer #3's comment #8, the development of high-performance stretchable TE materials and the first demonstration of intrinsically stretchable organic TE modules as well as their in-depth characterizations provide a number of important implications to the thermoelectric community including

- i) Importance of electromechanical properties of both TE legs and interconnects composing TE modules for stable energy harvesting under mechanical stretching (please refer our response to Reviewer #3's comment #5),
- ii) Independence of Seebeck coefficients from stretching conditions in the composites possessing excellent electromechanical properties (please refer our response to Reviewer #1's comment #2),
- iii) Strong correlation between Seebeck coefficients and electrical percolations in the composites supported by the suggested two DOS peaks model (please refer our response to Reviewer #3's comment #2),
- iv) Anisotropic behavior of the output power of stretchable TE modules under tensile strain in parallel and perpendicular directions (please refer our response to Reviewer #3's comment #6).

[Comment #2] What is the change in the Seebeck coefficients depending on the composition ratio of three substances, PEDOT:PSS, IL, and WPU?

[Response #2] To answer reviewer's comment, we measured Seebeck coefficient (S) of the composites over a full range of composition (i.e. different CP-WPU ratios, different contents of IL at optimized CP-WPU ratio, different types of IL) that we have characterized previously. The new results of S are now included along with the electrical and mechanical properties of the composites in Fig 2 & 3 in the revised manuscript. As shown in Fig. R5a, S of 21 $\mu\text{V}/\text{K}$ for the CP-TCM sample remains nearly the same with increasing the content of WPU (X) up to 85 (i.e. CP:WPU=(100- X): X =15:85). Note that the conductivity (σ_{dc}) of the composite decreases gradually with decreasing the content of CP up to this point. However, S starts to increase when $X > 90$ and reaches to $S=38 \mu\text{V}/\text{K}$ at $X=95$, while σ_{dc} decreases abruptly in this range. Strong correlation between S and σ_{dc} is also observed for the composites with different amount of IL (Fig. R5b). Here, the CP-WPU ratio is fixed, so a decrease in σ_{dc} with less amount of IL comes

from a reduced extent of electrical percolation. On the other hand, the amount of IL itself didn't affect S of CP-TCM samples without WPU (Fig R6a). Note that we measured S for all samples 3 or 4 days after the sample preparation due to the time dependence of S (Fig R6b) with the recovery of oxidation level over time (please refer our response to Reviewer #3's comment #4).

Figure R5. a) Electrical dc conductivity (σ_{dc}) and S of the composite with varying proportion of WPU (X) to CP (100-X). The composition ratio of the chosen IL (TCM) to CP was fixed at CP:TCM=15:25. b) σ_{dc} and S of the composite with varying content of TCM (Y). The proportion of WPU to CP was fixed at CP:WPU=15:85. Composition ratio, CP:TCM:WPU=15:Y:85.

Figure R6. a) S of CP films with varying content of TCM (Y). The relative content of CP is 15. 0.15 wt% of an ammonia solution was added with respect to the PEDOT:PSS dispersion as in the case for the composite with WPU. b) S of the optimized composite (CP:TCM:WPU=15:25:85, w/w) as a function of storage time in ambient conditions.

In order to explain this phenomenon, the increase in S with the reduced electrical percolation, we propose the following hypothesis. In good approximation, the transport of charge carriers in a hopping regime between localized states is governed by the density of state (DOS). The Fermi distribution dictates the filling of the DOS; which also determines a level of energy called the transport energy E^* (*J. Phys.: Condens. Matter* 1997, 9, 2699), dominating the percolation transport because of its highest hopping probability. The Seebeck coefficient (S) can be described as (*J. Appl. Phys.* **2003**, 93, 4653; *Phys. Rev. B* **2015**, 92,035201):

$$S \cong -\frac{E^* - E_F}{qT}$$

Where q is the charge of the carrier, T the temperature, E_F is the Fermi Level.

The density of percolation paths in PEDOT:PSS is strongly affected by the morphology of the film as known by the addition of (i) high boiling point solvents and ionic liquids that promotes the demixing of the excess of insulating PSS and promotes the creation of 3D percolation paths (*Adv. Electron. Mater.* 2015, 1, 1500017; *Adv. Mater.* 28, 8625 (2016)); or (ii) insulating polymers breaking the percolation paths (*Adv. Funct. Mater.* 2014, 24, 2957). Sometimes small amounts of those additives lead to a drastic change in transport properties. For instance, the same absorption spectrum is observed for films with an electrical conductivity varying by 3 orders of magnitude (*PNAS* 2018, 115, 11899), thus indicating that the concentration of PEDOT chains is barely affected, but their connectivity is much different and governed by microscopic morphology. Hence, we can thus speak about PEDOT chains with good connectivity and bad connectivity. PEDOT chains with good connectivity can form nanocrystals where short range order dominates the transport (blue domains in Fig. R7). While PEDOT chains with bad connectivity are in an amorphous phase without even local order (green domains). The electrostatic potential profile on those two different types of PEDOT chains is different (*Macromol. Rapid Commun.* 2018, 39, 1700533) and it is thus expected to affect their electronic energy levels. We propose that their DOS are thus also different. As a result, our hypothesis is to consider a DOS for PEDOT:PSS systems composed two DOS peaks. One DOS peak representing the electronic levels for the connected PEDOT chains and another DOS peak representing the electronic bipolaronic levels of the disconnected PEDOT chains. In this hypothesis, we thus

propose that upon the removal of ionic liquid or the addition of insulating WPU polymer, the ratio of connected vs. disconnected PEDOT chains decreases and the intensity ratio of those two DOS peaks differs, which leads to an increase of S as proposed recently for polymer blends (*Adv. Electron. Mater.* 2019, 5, 1800821).

Figure R7. Schematic drawings of different regimes of PEDOT connectivity (bottom) and the corresponding electronic energy structures (top).

This hypothesis is now included in the revised Supplementary Information. We revised our manuscript in accordance with new results and a hypothesis as shown below.

(on page 7) Up to this point, the S of the composite is nearly identical ($S \approx 22 \mu\text{V/K}$), but further loading of WPU increased S to $38 \mu\text{V/K}$ at WPU:CP=95:5.

(on page 8) As the electrical percolation diminishes with lower amounts of IL, the S increases inversely. Note that the amount of IL has no effect on the S of CP-TCM composites without

WPU (see Supplementary Fig. S1). A potential hypothesis to understand the correlation of S with percolation is proposed in Supplementary Information (Fig. S2).

(on page 19) Note that we measured S for all samples 3 or 4 days after the sample preparation due to the time dependence of S with the recovery of oxidation level over time (see Supplementary Fig. 1b).

[Comment #3] On line 206, the author referred to a paper that is not yet published.

[Response #3] We thank the reviewer's correction. Since this paper is still under consideration of publication, we removed this reference and explained the reason of adding ammonia in more detail in the revised manuscript as shown below.

(on page 10) Since we added a small amount of ammonia (CP:NH₃=8:1) in all composite dispersions to prevent acid-induced aggregation of anionic WPU by protonation from PSSH,

[Comment #4] On line 209, the author claims that the conductivity of composite decreased by dedoping is recovered via chemical equilibrium with the oxygen in air. However, no evidence or reference can be found to support the author's claim in the manuscript. Provide the appropriate evidences or refer relevant papers.

[Response #4] To provide evidences for dedoping of PEDOT by adding ammonia (NH₃) into the PEDOT:PSS dispersion and recovery of doping level in air after film formation, we measured absorbance spectra of CP-TCM films as a function of storage time. It is well-known that optical transitions observed in absorption spectra reflect electronic states governed by doping level (*J. Mater. Chem. C* 2014, 2, 1278). Doped PEDOT shows two main absorptions: a broad free carrier absorption in NIR region and a polaron absorption peak at around 900 nm. Note that these absorption features are not clear in the spectra of CP-TCM-WPU films due to the absorption of WPU (*J. Mater. Chem. A*, 2018,6, 9192). As shown in Fig. R8, the CP-TCM film coated from the

dispersion containing NH_3 (CP: NH_3 =8:1) initially shows a pronounced polaron peak at around 900 nm and a reduced free carrier absorption in the NIR region as a result of partial dedoping of PEDOT occurring in the relatively basic dispersion compared with the pristine PEDOT:PSS dispersion (*J. Mater. Chem. C*, 2015, 3, 10616; *Int. J. Photoenergy*, vol. 2012, Article ID 598903). As the CP-TCM- NH_3 film was stored in air, the NIR absorption increased with a decrease in the polaron peak and the spectrum became identical with that of the CP-TCM film coated from the dispersion not containing NH_3 after 25 hours of storage in air. It has been proven that electrochemically reduced PEDOT is oxidized spontaneously upon oxygen exposure in the dry state (*J. Mater. Chem. A*, 2017, 5, 4404). In response to the reviewer's comment, we added this new data in the Supplementary Information as evidences and provided references in the revised manuscript as shown below.

(on page 10) the initial σ_{dc} was lower due to the partial dedoping of the PEDOT occurring in the relatively basic dispersion containing NH_3 (see Supplementary Fig. S7a,c)^{36,37}. The increase in σ_{dc} of the composite over time is attributed to the recovery of the doping level via spontaneous oxidation upon oxygen (air) exposure³⁸ as evidenced by the changes in absorption spectra (see Supplementary Fig. S7c).

(on page 8 in Supplementary) The CP-TCM film coated from the dispersion containing NH_3 initially shows a pronounced polaron peak at around 900 nm and a reduced free carrier absorption in the NIR region¹⁰ as a result of partial dedoping of PEDOT. As the CP-TCM- NH_3 film was stored in air, the NIR absorption increased with a decrease in the polaron peak and the spectrum became identical with that of the CP-TCM film coated from the dispersion not containing NH_3 after 25 hours of storage in air. Composition ratio, CP:TCM: NH_3 =12:20:1.5 or CP:TCM=12:20, w/w.

Figure R8. Comparison of absorbance spectra of CP-TCM films over time. The films were stored in ambient air. Composition ratio, CP:TCM:NH₃=12:20:1.5 or CP:TCM=12:20, w/w.

[Comment #5] On line 318, are the Au-TiO₂ NWs stretchable? (If they are stretchable enough, please provide the proper evidence.) Resistance of the Au-TiO₂ NWs is significantly increased with an increase of tensile strain in Figure S15. I wonder the drop in the output power of the TE module in Figure 5f is closely related to increased resistance of the Au-TiO₂ NWs.

[Response #5] The Au-TiO₂ NWs embedded in PDMS substrates can be called “stretchable conductors” because this material is electrically conductive upon repeated tensile stretching above 100% owing to the retained connection between conductive NWs even under large deformations (see Fig. S18a in the revised Supplementary and ref. *Adv. Mater.* 2018, 1706520). However, the increase in R of Au-TiO₂ NWs when stretched is relatively large compared with the increase observed for the composite. So, the reviewer’s insight is correct, and we agree that it is important to address explicitly the main cause of R_{in} (total internal resistance of TE module) increase that proportionally reduces the output power of the TE module. Equivalent circuits drawn in Fig. R9 (Fig. S19 in the revised Supplementary Information) shows that R_{in} is the summation of resistances of a TE leg (R_{leg}), an Au-TiO₂ NW interconnect (R_{Au}), and a contact resistance between a leg and an interconnect (R_c). Comparing the changes in R_{in} and ΣR_{Au} upon

tensile stretching reveals that the R_{in} increase arises mainly from the increase in ΣR_{Au} , while the change in $\Sigma R_{leg} + \Sigma R_c$ estimated by subtracting ΣR_{Au} from R_{in} has minor contribution to the R_{in} increase. In accordance with our response to the next comment, we did the same analysis on the TE module stretched in the direction perpendicular to thermodiffusion. It is interesting to see that ΣR_{Au} increases upon perpendicular stretching even though it is expected to decrease slightly based on geometric changes (i.e. decrease in length and thickness, increase in width, while $R \propto \text{length}/(\text{width} \times \text{thickness})$), presumably due to the reduced density of connections between conductive NWs. A slight increase in $\Sigma R_{leg} + \Sigma R_c$ can be understood in the same manner for CP nanofibrils. This analysis is included in the revised Supplementary Information with the revision of main text as shown in our response to Reviewer #3's comment #6.

Figure R9. a,b) Schematics of TE modules characterized in Fig. 6 under tensile strain in the direction (a) parallel with and (b) perpendicular to thermodiffusion. Equivalent circuits to calculate the internal resistance of TE modules (R_{in}) were drawn below. $R_{in} = \sum R_{leg} + \sum R_{Au} + \sum R_c$, where R_{leg} , R_{Au} , and R_c are the resistances of a TE leg, an Au-TiO₂ NW interconnect, and a contact between a leg and an interconnect, respectively. c,d) Contributions of $\sum R_{Au}$ and $\sum R_{leg} + \sum R_c$ to R_{in} . $\sum R_{Au}$ was estimated by multiplying R of a single interconnect measured under different

tensile strain and the number of interconnects in each module (i.e. 9 for (c) and 4 for (d)). R_{Au} vs. strain (ϵ) in (c) was obtained on a motorized linear stage with gold-coated 4-point contact pads, while R_{Au} vs. ϵ in (d) was measured in the same way as for the TE module by the 2-point probe method. $\Sigma R_{leg} + \Sigma R_c$ vs. ϵ was estimated by subtracting ΣR_{Au} from R_{in} . Comparing the changes in R_{in} and ΣR_{Au} reveals that the R_{in} increase arose mainly from the increase in ΣR_{Au} , upon both parallel and perpendicular stretching, while the change in $\Sigma R_{leg} + \Sigma R_c$ has minor contribution to the R_{in} increase. Despite a reduced length of interconnects upon perpendicular stretching, the ΣR_{Au} increase can be caused by the reduced density of connections between conductive NWs.

[Comment #6] In page 30, the authors evaluated the TE properties of the stretchable TE module in only one direction. Since the active part of the TE module shown in Figure 5 has a long rectangular shape with anisotropy (not a square), they should be characterized in both perpendicular and parallel direction to elongation.

[Response #6] In response to Reviewer #3's comment, we characterized the performance of the TE module upon tensile stretching in the direction perpendicular to thermodiffusion as shown in Fig. R10 (Fig. 6 in the revised manuscript). As we have already discussed in our response to Reviewer #3's comment #5, R_{in} was increased even by perpendicular stretching, but the increase was much less than in the case of parallel stretching (Fig. R10b, e) so that 83% of the initial $P_{out-max}$ is maintained under 40% strain (Fig. R10f). Note that we couldn't help but reduce the later size of the module, i.e. the number of TE legs in the module, for a practical reason that the Peltier elements equipped in our set-up have a limited size and can't cover 10-legs modules when stretched to 40% in a perpendicular direction. Accordingly, we revised our manuscript as shown below.

(on page 15) The tensile loading up to 40% in both parallel and perpendicular directions does not affect the V_{OC} of the stretchable module (Fig. 6e,h). However, the R_{in} increases gradually with increasing strain due to the strain-dependent resistances of the composite legs and the interconnects (see Supplementary Fig. S18). The analysis of the strain-dependent resistances with equivalent circuits proves that the R_{in} increases arose mainly from increased resistance of the

Au-TiO₂ NW interconnects for both stretching directions, while the R_{in} increase was much less pronounced for perpendicular strain (Fig. 6e,h and Supplementary Fig. S19). According to the relation $P_{out} \propto 1/R_{in}$, the $P_{out-max}$ gradually decreases with increasing strain, with 48% and 83% of the initial $P_{out-max}$ maintained for 40% parallel and perpendicular strain, respectively (Fig. 6f, 6i).

Figure R10. a-c) TE module stretched in the direction parallel with thermodiffusion. $T_1=23$ °C, $T_2=53$ °C. (b) V_{OC} and R_{in} vs. tensile strain (ϵ) applied to a module for $\Delta T=30$ K. (c) P_{out} vs. R_{load} under different strains for $\Delta T=30$ K. d-f) TE module stretched in the direction perpendicular to thermodiffusion and corresponding V_{OC} and R_{in} vs. ϵ and P_{out} vs. R_{load} .

[Comment #7] On line 650, the elasticity of TE module with the composite is different from that of the pristine active material because the TE module includes both substrate and electrode. Is there no effect of the substrate and the electrode on the performance of the active layer? Also, direction of stretching should be indicated with an arrow in Figure 5 for the reader's understanding.

[Response #7] Fig. R11 (Fig. S18b in the revised Supplementary Information) shows that the electromechanical property of the free-standing composite is different from the composite supported on a PDMS substrate. As the effect of supporting substrates is also shown for the electromechanical property under cyclic stretching (see Fig. R1 and our response to Reviewer #1's comment #2), we assume that different electromechanical properties originate from different mechanical deformations between the composite and a PDMS substrate since the composite ($\sim 40 \mu\text{m}$) supported on a PDMS substrate conforms to the substrate ($>200 \mu\text{m}$) when deformed. Nonetheless, the difference is not significant under tensile strain up to 40%. Meanwhile, embedding Au-TiO₂ NWs ($<5 \mu\text{m}$) in a PDMS substrate has a negligible effect on the mechanical property of the substrate as proven in the reference (*Adv. Mater.* 2018, 1706520), so neither on that of the composite supported on a PDMS. To clarify this, we added a new result in the revised Supplementary Information as shown below. Also, as suggested by Reviewer #3, we indicated the direction of stretching TE modules as well as the schematics of geometrical changes in TE modules in Fig. 6 in the revised manuscript.

Figure R11. Resistance of a TE leg (i.e. the optimized composite film supported on a PDMS substrate, gray line) as a function of tensile strain with the comparison to the free-standing optimized composite film (data from Fig. 3b, blue line). Different electromechanical properties

with and without PDMS support can originate from different mechanical deformations between the composite and a PDMS substrate since the composite ($\sim 40 \mu\text{m}$) supported on a PDMS substrate conforms to the substrate ($>200 \mu\text{m}$) when deformed. Nonetheless, the difference is not significant under tensile strain up to 40%.

[Comment #8] On line 305, for wearable and stretchable TE modules, vertical configuration is more suitable than parallel one due to its easy build-up of the temperature gradient and low electrical resistance. If the PEDOT:PSS-IL-WPU composite forms the enough thickness, vertical TE module can be prepared by using only p-type TE legs. Is the thickness of the PEDOT:PSS-IL-WPU films ($\sim 40 \mu\text{m}$) sufficient for the TE legs of vertical configuration? If it is impossible, how can the thickness be increased or controlled? How can the electrical conductivity and mechanical properties be optimized depending on thickness?

[Response #8] As we have also discussed briefly in the section following after line 305, we strongly agree with Reviewer #3 that vertical configuration is more suitable for wearable TE modules, especially to maximize the output power harvested from our body heat due to its geometry that may reduce R_{in} (i.e. enhance P_{out}) by 3-4 orders of magnitude compared with lateral configuration. The current thickness of our composite ($\sim 40 \mu\text{m}$) can already be used as the TE legs of vertical configuration in case of micro-TE generators (*J. Microelectromech. S.* vol. 27, no. 1, 1, 2018). But, the thickness of our composite can be increased simply by increasing the drop-casting volume of composite inks. Nonetheless, as we have discussed regarding Reviewer #3's comment #5, not only stretchable TE legs but also stretchable interconnects are important elements to build stretchable TE modules, and the demands on both materials and processes for interconnects would be very steep to build vertical TE modules without comparable *n*-type TE materials. Therefore, we think that this issue is out of scope for this work but is important and should be considered for the next project.

REVIEWERS' COMMENTS:

Reviewer #1 (Remarks to the Author):

I appreciate the effort the authors expended into fully addressing the reviewer comments. I believe this manuscript not only represents a significant advancement in the field of flexible thermoelectrics, but also contains sufficient detail and corollary data to motivate and guide the research efforts of the broad flexible electronics community and is therefore suitable for publication in its current form.

Reviewer #3 (Remarks to the Author):

In this revised manuscript, experimental results are systematically demonstrated to address a relationship between Seebeck coefficient and electrical conductivity of elastic conducting polymers with IL and WPU. Also, new investigation is suggested for the analysis on anisotropy of a thermoelectric module for stretchable electronics and effect of Au-TiO₂ NW electrodes on thermoelectric performance with relevant discussions. This revised manuscript is significantly improved and I believe this manuscript is sufficient for publication in Nature Communication.

REVIEWERS' COMMENTS:

Reviewer #1 (Remarks to the Author):

The authors report the fabrication of a novel PEDOT-ionic liquid composite that exhibits high conductivity, elasticity, stretchability, and a low young's modulus. The authors attribute the high performance of their material to favorable nano- and micro- scale percolation networks, and plasticizing effect of the ionic liquid, and then use this material to fabricate an intrinsically stretchable organic thermoelectric module. To support their hypothesis of the importance of percolation networks, the authors investigated the properties of their composite over a wide range of composition, and measured the size distribution of aggregates in the aqueous dispersions of the composite that they used to fabricate films of their materials, as well as performed atomic force microscopy measurements to visualize the surface morphology of their films.

Although intrinsically stretchable thermoelectric materials have been previously demonstrated, the fabrication of a thermoelectric module with tens of micron thick legs from such a material, along with the increased electrical conductivity and supported hypothesis for this increase, represents a significant advance for the understanding and utility of these materials. This manuscript is both sufficiently novel, and relevant to a broad audience to justify publication in Nature Communications after addressing the following comments:

1)

Although the authors report electrical properties as a function of strain and reference cycling tests for the physical properties of their materials, it is unclear how the electrical properties evolve with strain cycling. If the conducting polymer is formed of a percolated network of nanofibrils, it seems reasonable that cycling would induce changes in this morphology that would affect the electrical conductivity or Seebeck coefficient. Additionally, although electrical conductivity can be inferred from the change in resistance and geometric changes for strain up to 100%, the Seebeck coefficient is shown only up to strains of 40%. If the authors have additional data on electrical properties as a function of cycle, and Seebeck coefficients for larger strain values, these data would be useful for a reader to understand the importance of the demonstrated thermoelectric module.

2)

Similar to above, although the resistance has increased and therefore current output of the thermoelectric module decreased by a factor of approximately two for tensile strains of 40%, these changes are explained by geometric changes in the material only if it behaves as a material whose resistivity is an intensive material property. Given that percolation through a network of fibers is a more complex model of conduction, the authors should comment on the validity of their assumptions when using an effective medium model to determine the conductivity of the embedded conducting polymer phase, or uniform geometric model that invokes intensive material properties, such as for 2(h) and S3.

3)

The DC conductivities were measured on spun-cast films that were significantly thinner than the legs of the drop-cast thermoelectric modules. The authors should comment on potential differences in morphologies of materials deposited in these ways, and specifically how those differences might affect a percolation model of conduction.

Reviewer #2 (Remarks to the Author):

This is an interesting paper on the development of flexible thermoelectric based on PEDOT. Inclusion of ionic liquids (ILs) into the material make possible the development of stretchable thermoelectrics. The work has been carried out very well and the results should be of interest to the thermoelectric community. Unfortunately, while I think the work is good, I don't believe it has the level of impact and novelty that one would expect for the journal. I don't believe it will impact on thinking in the field to a sufficient level.

On a scientific level, I note that the authors haven't really addressed loss or leaching of the plasticiser from the materials. In the development of composite materials based on ILs often must determine the stability, dispersion and loss of the material in/from the host material. I think this is an important issue that must be addressed.

Reviewer #3 (Remarks to the Author):

The authors report on the stretchable PEDOT:PSS-IL-WPU composite for the stretchable thermoelectric (TE) module. In this manuscript, the solution processable TE ink is prepared using the PEDOT:PSS-IL-WPU composite and this composite exhibits quite good electrical and mechanical properties. However, materials based on PEDOT:PSS with elastomer have widely been studied even with their use in stretchable electronics. Furthermore, the addition of IL for the TE application of PEDOT:PSS is a routine process. The fabricated TE module in parallel direction without n-type materials also shows quite low output power despite the good mechanical stability.

Although the paper is well written and presented in a scholarly manner with few exceptions, I believe this manuscript, while interesting, is insufficient to provide readers with useful information in the field of thermoelectronics. Therefore, I strongly recommend rejection for publication in Nature Communications.

- 1) What is the change in the Seebeck coefficients depending on the composition ratio of three substances, PEDOT:PSS, IL, and WPU?
- 2) On line 206, the author referred to a paper that is not yet published.

3) On line 209, the author claims that the conductivity of composite decreased by dedoping is recovered via chemical equilibrium with the oxygen in air. However, no evidence or reference can be found to support the author's claim in the manuscript. Provide the appropriate evidences or refer relevant papers.

4) On line 318, are the Au-TiO₂ NWs stretchable? (If they are stretchable enough, please provide the proper evidence.) Resistance of the Au-TiO₂ NWs is significantly increased with an increase of tensile strain in Figure S15. I wonder the drop in the output power of the TE module in Figure 5f is closely related to increased resistance of the Au-TiO₂ NWs.

5) In page 30, the authors evaluated the TE properties of the stretchable TE module in only one direction. Since the active part of the TE module shown in Figure 5 has a long rectangular shape with anisotropy (not a square), they should be characterized in both perpendicular and parallel direction to elongation.

6) On line 650, the elasticity of TE module with the composite is different from that of the pristine active material because the TE module includes both substrate and electrode. Is there no effect of the substrate and the electrode on the performance of the active layer? Also, direction of stretching should be indicated with an arrow in Figure 5 for the reader's understanding.

7) On line 305, for wearable and stretchable TE modules, vertical configuration is more suitable than parallel one due to its easy build-up of the temperature gradient and low electrical resistance. If the PEDOT:PSS-IL-WPU composite forms the enough thickness, vertical TE module can be prepared by using only p-type TE legs. Is the thickness of the PEDOT:PSS-IL-WPU films (~40 μm) sufficient for the TE legs of vertical configuration? If it is impossible, how can the thickness be increased or controlled? How can the electrical conductivity and mechanical properties be optimized depending on thickness?

Responses to Reviewer #1's comments:

[Comment #1] The authors report the fabrication of a novel PEDOT-ionic liquid composite that exhibits high conductivity, elasticity, stretchability, and a low young's modulus. The authors attribute the high performance of their material to favorable nano- and micro- scale percolation networks, and plasticizing effect of the ionic liquid, and then use this material to fabricate an intrinsically stretchable organic thermoelectric module. To support their hypothesis of the importance of percolation networks, the authors investigated the properties of their composite over a wide range of composition, and measured the size distribution of aggregates in the aqueous dispersions of the composite that they used to fabricate films of their materials, as well as performed atomic force microscopy measurements to visualize the surface morphology of their films.

Although intrinsically stretchable thermoelectric materials have been previously demonstrated, the fabrication of a thermoelectric module with tens of micron thick legs from such a material, along with the increased electrical conductivity and supported hypothesis for this increase, represents a significant advance for the understanding and utility of these materials. This manuscript is both sufficiently novel, and relevant to a broad audience to justify publication in Nature Communications after addressing the following comments:

[Response #1] We thank Reviewer #1 for evaluating our paper and appreciating the novelty and significance of our work on developing highly conductive elastic conducting polymer composites and demonstrating the first intrinsically stretchable organic thermoelectric modules based on the composites. As Reviewer #1 has pointed out, intrinsically stretchable thermoelectric materials based on PEDOT:PSS/elastomer composites and PEDOT:PSS/plasticizer composites, which we have also discussed in detail in the introduction section, have been previously reported. However, none of those materials could constitute the real module applications due to low conductivity of elastomeric composites or poor mechanical property of plasticizer composites as well as the lack of suitable stretchable interconnects. So, we agree with Reviewer #1 that our work represents a significant advance in the field of stretchable thermoelectrics. We tried our best to address all the valuable comments from Reviewer #1.

[Comment #2] Although the authors report electrical properties as a function of strain and reference cycling tests for the physical properties of their materials, it is unclear how the electrical properties evolve with strain cycling. If the conducting polymer is formed of a percolated network of nanofibrils, it seems reasonable that cycling would induce changes in this morphology that would affect the electrical conductivity or Seebeck coefficient. Additionally, although electrical conductivity can be inferred from the change in resistance and geometric changes for strain up to 100%, the Seebeck coefficient is shown only up to strains of 40%. If the authors have additional data on electrical properties as a function of cycle, and Seebeck coefficients for larger strain values, these data would be useful for a reader to understand the importance of the demonstrated thermoelectric module.

[Response #2] According to the Reviewer #1's comment, we examined the electrical properties of the optimized composites (CP:TCM:WPU=15:25:85, w/w) as a function of stretching cycle at three different tensile strains (20%, 40%, and 60%). As shown in Fig. R1 (Fig. 3e in the revised manuscript), the composite shows excellent cycling stability when stretched to 20% and 40% 1000 times. The initial increase in resistance (R) caused by the separation between CP nanofibrils with geometric changes declined gradually over cycles presumably due to the alignment and reconnection of nanofibrils (*Sci. Adv.* 2017; 3 : e1602076). Similar behavior was also observed for AuNW network (*Adv. Mater.* 2018, 1706520). R of the composite became even less than the original R when released after 1000 cycles at 20% or increased by only 6% after 1000 cycles at 40%. However, further rearrangement of conductive fibrils under 60% strain caused plastic deformation in order to accompany the stress focused on the weak points of the composite (e.g. relatively thinner parts) and the repeated stress resulted in the complete tear of these points after approximately 200 cycles (see Fig. R2). Nonetheless, when the composite was supported on a PDMS substrate as in the case for device applications, the stress is applied more uniformly across the substrate and the process for plastic deformation is much slowed down. As a result, the supported composite could withstand 1000 stretching cycles at 60% strain with the increase in R by 70%.

Figure R1. Resistance changes (R/R_0) of the optimized composite (CP:TCM:WPU=15:25:85) upon cyclic tensile stretching (blue lines). R/R_0 of the optimized composite supported on a PDMS substrate under cyclic strain at 60% is also shown (gray line).

Figure R2. a) Resistance changes (R/R_0) of the free-standing optimized composite (CP:TCM:WPU=15:25:85) under cyclic strain at 60%. b) Optical microscope images of the torn parts of the composite after 200 cycles.

Seebeck coefficients of the optimized composite for larger strain values up to 100% were also examined as shown in Fig. R3 (Fig. 3c in the revised manuscript). Seebeck coefficients were retained nearly the same under strains up to 80% and slightly decreased at 100% strain, indicating that the density of percolation paths and the corresponding density of states are maintained in this strain range (*Adv. Electron. Mater.* 2019, 5, 1800918 & *Nat. Mater.* 13, 190-194 (2014), and please refer our response to Reviewer #3's comment #2).

Figure R3. Seebeck coefficients (S) of the composites with a different type of ILs and S vs. tensile strain (ϵ) for the optimized composite.

In response to Reviewer #1's comment, we included the new results in the manuscript and supplementary information and revised the main text as shown below.

(on page 9) S does not vary much depending on the type of ILs, while the inverse relationship between σ_{dc} and S of the composite still holds with different ILs that cause different percolative networks. The S of the composite with TCM was nearly constant for strains up to 80% and decreased slightly at 100% strain.

(on page 9) The R/R_0 of the composite with TCM shows excellent cycling stability when stretched up to 40% (Fig. 3e). R became even less than R_0 when released after 1000 stretching cycles at 20% or increased by only 6% after 1000 cycles at 40%. The decline in R over cycles is ascribed to the alignment of CP fibrils¹⁸. However, the viscoelastic property of the composite caused partial plastic deformation at 60% strain and repeated stresses focused on weak points resulted in the complete tear after approximately 200 cycles (see Supplementary Fig. S6). Nonetheless, when supported on a PDMS substrate, more uniformly distributed stresses across the substrate suppress plastic deformation so that the composite could withstand 1000 stretching

cycles at 60% with a R/R_0 of only 1.7.

[Comment #3] Similar to above, although the resistance has increased and therefore current output of the thermoelectric module decreased by a factor of approximately two for tensile strains of 40%, these changes are explained by geometric changes in the material only if it behaves as a material whose resistivity is an intensive material property. Given that percolation through a network of fibers is a more complex model of conduction, the authors should comment on the validity of their assumptions when using an effective medium model to determine the conductivity of the embedded conducting polymer phase, or uniform geometric model that invokes intensive material properties, such as for 2(h) and S3.

[Response #3] The reviewer is correct that the complex geometry of the CP phase within the composite is unknown. However, by considering the orientation factor (conduction pathways are not aligned in the macroscopic direction of conduction) from an ideally connected geometry, e.g. the fiber contact model, we find that the effective conductivity of the CP phase is approximately equal to that of pure CP thin films. This would not be the case if the CP phase constituted a poorly percolating network, as in that case a lot of the CP material would not contribute to the conduction. Thus, the analysis indicates that our CP network exhibits close to ideal percolation behavior. We have clarified the assumptions for this analysis in the text.

(on page 10) Due to the orientation of the conducting segments within a composite, the σ_{dc} is typically lower in a composite than in a pure conductor. (*Int. J. Hydrogen Energy* 42, 9262 (2017)) Even for well-connected systems, like meshes of randomly oriented nanowires or connected tubes of 45° angle with respect to the direction of conduction (fiber contact model), the σ_{dc} is reduced by 50 %. Although the exact geometry of the CP phase in our composite is unknown, by assuming that it possesses the same orientational characteristics as the fiber geometries described above, the effective σ_{dc} for the CP phase would be approximately 2340 S cm^{-1} . This is nearly the same value as for the CP film without WPU (≈ 2350 S cm^{-1} , see Supplementary Fig. S3), indicating that most of the PEDOT network within the composite is contributing to the conduction.

To avoid possible misleading, we showed in Fig. 3b and Fig. S5 (Fig. 2h and Fig. S3 in the previous version) the changes in R and geometry of the composites as a function of strain, which can be directly measured, rather than showing the conductivity change that can be estimated from R and geometric changes with large errors. The R_G/R_0 (*exp.*) values plotted in Fig. 3b come from simple mathematics ($R \propto \text{length}/(\text{width} \times \text{thickness})$) based on average values for real geometric changes of the composite shown in Fig. S5 without employing any geometric model with the comparison to the case for rubber having a Poisson's ratio of 0.5. To clarify this, we revised the text in the Supplementary Information as shown below.

(on page 7 in Supplementary) R_G/R_0 (*exp.*) in Fig. 3b was estimated from the average values measured for dw/w_0 and dt/t_0 plotted in Fig. S5 according to the relation:

$$\frac{R_G}{R_0} = \frac{1 + dl/l_0}{(1 + dw/w_0)(1 + dt/t_0)}$$

[Comment #4] The DC conductivities were measured on spun-cast films that were significantly thinner than the legs of the drop-cast thermoelectric modules. The authors should comment on potential differences in morphologies of materials deposited in these ways, and specifically how those differences might affect a percolation model of conduction.

[Response #4] Spin-coating is a useful technique to form thin ($<1 \mu\text{m}$) films having uniform thicknesses. Centrifugal forces that uniformly spread the dispersion across the substrate as well as fast drying ensure the formation of homogeneous films, while centrifugal forces often induces the orientation of polymer molecules that enhances the charge carrier mobility (*Journal of Nanomaterials*, Volume 2017, Article ID 3624750, 18 pages). In contrast, drop-casting takes much longer drying time with differences in drying rates across the substrate. Therefore, there could be concentration gradients or more phase separation (*ACS Appl. Mater. Interfaces* 2015, 7, 35, 19764) between different components that results in less homogeneous films with larger variations in thickness compared to spin-coated films. Accordingly, drop-cast films ($>10 \mu\text{m}$) usually show lower conductivity values than spin-coated films of the same materials. For instance, the conductivity values reported for the most representative PEDOT:PSS formulations (i.e.

PEDOT:PSS with 5 wt% of DMSO) were different from drop-cast films (≈ 660 S/cm, *Phys. Rev. Lett.* 109, 106405 (2012)) and spin-coated films (≈ 890 S/cm, *ACS Appl. Mater. Interfaces* 2016, 8, 11629) by 1.35 fold. Our composite (CP:TCM:WPU=15:25:85) also shows lower conductivity for the drop-cast films (≈ 100 S/cm) used as thermoelectric legs than the spin-coated films (≈ 140 S/cm,) by 1.4 fold. Based on the similar difference in conductivities observed for the representative PEDOT:PSS formulations, we assume that the effect of the different deposition methods is related to morphological differences induced by the film formation process. Note however that the conductivity stays in the same order of magnitude, which is a clear indication that the microscopic morphology of the CP governing the charge transport in the insulating matrix is not completely different between spin-coated and drop-cast films. To clarify this, we revised our manuscript as shown below.

(on page 10) Compared with the spin-coated film ($\sigma_{dc} \approx 140$ S cm⁻¹), the drop-cast film of the optimized composite has only a slightly lower σ_{dc} (≈ 100 S cm⁻¹), a similar reduction has been observed for CP-DMSO films^{41,42}; which indicates that thicker free-standing drop-cast films used below in thermoelectric generators have likely similar transport mechanism and microscopic morphology as spin-coated films.

Responses to Reviewer #2's comments:

[Comment #1] This is an interesting paper on the development of flexible thermoelectric based on PEDOT. Inclusion of ionic liquids (ILs) into the material make possible the development of stretchable thermoelectrics. The work has been carried out very well and the results should be of interest to the thermoelectric community. Unfortunately, while I think the work is good, I don't believe it has the level of impact and novelty that one would expect for the journal. I don't believe it will impact on thinking in the field to a sufficient level.

[Response #1] We thank Reviewer #2 for evaluating our paper and appreciating the excellence of our work. Along with the recent paradigm shift from flexible electronics toward stretchable electronics, an appealing challenge is to realize intrinsically stretchable thermoelectric modules that can be attached conformably to our body to power wearable electronics using body heat. Despite increasing efforts to develop intrinsically stretchable thermoelectric materials based on organic materials focusing on PEDOT, the failure to simultaneously achieve both good thermoelectric properties (electrical conductivity, Seebeck coefficient) and excellent mechanical properties (stretchability, elasticity, softness) in the form thick enough to be used in thermoelectric modules has been limiting the demonstration of intrinsically stretchable organic thermoelectric modules. Instead, stretchable thermoelectric generators have been recently demonstrated based on deformable coil structures or tearing processes of rigid inorganic materials with the limitations such as out-of-plane deformation or severe degradation of output power upon stretching (*Sci. Adv.* 2018, 4 : eaau5849; *Energy Environ. Sci.* 2016, 9, 1696). In our work, for the first time, we demonstrated truly intrinsically stretchable organic thermoelectric modules by developing a novel composite material that simultaneously possesses high electrical conductivity, high stretchability, and elasticity with low Young's modulus in the thick free-standing form, properties which never have been achieved simultaneously in any organic conductor previously. Our novel composite material enables not only thermoelectric generators but also other PEDOT-based devices requiring thick films (supercapacitors, actuators, biosensors, etc.) to be used in mechanically demanding conditions. Moreover, the presented strategy to successfully combine a number of properties in a single composite material and the implication revealed by characterizing the first demonstrated intrinsically stretchable organic thermoelectric

modules (please refer our response to Reviewer #3's comment #1 for details) would have significant impacts to design a novel class of nanocomposites and for the further development of intrinsically stretchable thermoelectric modules. Therefore, we believe that our work will be of interest to the broad readership in the fields of material/polymer science, organic electronics and thermoelectrics.

[Comment #2] On a scientific level, I note that the authors haven't really addressed loss or leaching of the plasticiser from the materials. In the development of composite materials based on ILs often must determine the stability, dispersion and loss of the material in/from the host material. I think this is an important issue that must be addressed.

[Response #2] We appreciate the Reviewer #2's valuable comment. The stability of the material upon stretching cycles is discussed in the response #2 of Reviewer #1. Here, we focus on the stability of the material with time and temperature. As the reviewer concerned, loss of plasticizers is an important issue that can deteriorate the physical properties of composites. Although the negligible volatility of ionic liquids compared with traditional plasticizers (*European Polymer Journal* 39 (2003) 1947) solves this issue of loss of plasticizers through evaporation, migration of plasticizers inside or out of the host material still can lead to loss of flexibility caused by the reduced dispersion or leaching of plasticizers. To assess the stability of our composite, we examined optical microscope images and stress-strain curves of the composites (CP:TCM:WPU=15:25:85) as a function of storage time and annealing temperature. As shown in Fig. R4, there is no notable change in the surface morphology over time up to 7 months except the creation of tiny lumps on the surface. Meanwhile, the 7-months-old composite retains its initial conductance (e.g. R of the sample measured 7 months after the sample preparation is nearly the same as that measured 3 days after the sample preparation) and possesses even higher elongation at break than the fresh sample. Therefore, we assume that tiny lumps may originate from small-molecular polyurethanes that migrated from the bulk to the surface over time, while this migration would not have the effect on the electrical percolation or the mechanical connection between large-molecular polyurethanes. The composite shows also excellent stability against thermal annealing at 80 °C as proven by the similar morphology and mechanical property to the sample before thermal annealing. However, thermal annealing at 100 °C led to significantly

different morphology in optical microscope images in conjunction with a severe deterioration in its mechanical property (i.e. elongation at break decreased down to 27%). Based on haze in the image and the poor mechanical property observed for the composite without ionic liquids, we assume that the phase separation between different components and the migration of ionic liquids to the surface may occur at this temperature, implying that thermoelectric modules made of our composites are not suitable for high temperature operation above 100 °C. To address this issue clearly, we revised the manuscript and supplementary information as shown below.

(On page 15) After 7 months of storage in ambient conditions, the TE module retained the same R_{in} and 86% of the original V_{OC} (see Supplementary Fig. S1b) owing to the excellent stability of the CP-TCM-WPU composite (see Supplementary Fig. S20) and the Au-TiO₂ NWs⁴⁹. However, thermal annealing at temperature above 100 °C led to a severe deterioration in the mechanical properties of the composite, possibly by induced phase separation and the migration of ILs, indicating that the TE module is not suitable for high temperature operation.

(on page 20 in Supplementary) The similar surface morphology and mechanical property preserved after 7 months storage time or thermal annealing at 80 °C show the excellent stability of the composite with a stable dispersion of ionic liquid plasticizers. Considering the preserved electrical and mechanical properties over time, the tiny lumps observed on the surface of old samples may result from the migration of small-molecular polyurethanes. Thermal annealing at 100 °C led to a significantly different morphology in conjunction with a severe deterioration in mechanical property (i.e. elongation at break decreased down to 27%). Based on haze in the image and poor mechanical property observed for the composite without ionic liquids, we assume that the phase separation between different components and the migration of ionic liquids to the surface may occur at this temperature.

Figure R4. Optical microscope images and stress (σ)-strain (ε) curves of the optimized composite (CP:TCM:WPU=15:25:85, w/w) films used for TE legs after storage time up to 7 months in ambient conditions or after heating up to 100 °C.

Responses to Reviewer #3's comments:

[Comment #1] The authors report on the stretchable PEDOT:PSS-IL-WPU composite for the stretchable thermoelectric (TE) module. In this manuscript, the solution processable TE ink is prepared using the PEDOT:PSS-IL-WPU composite and this composite exhibits quite good electrical and mechanical properties. However, materials based on PEDOT:PSS with elastomer have widely been studied even with their use in stretchable electronics. Furthermore, the addition of IL for the TE application of PEDOT:PSS is a routine process. The fabricated TE module in parallel direction without n-type materials also shows quite low output power despite the good mechanical stability.

Although the paper is well written and presented in a scholarly manner with few exceptions, I believe this manuscript, while interesting, is insufficient to provide readers with useful information in the field of thermoelectronics. Therefore, I strongly recommend rejection for publication in Nature Communications.

[Response #1] We thank Reviewer #3 for evaluating our paper and constructive comments that were helpful to greatly improve the scientific level and impact of our paper. As the reviewer pointed out and also as we have already discussed in the introduction of our manuscript, the composites of PEDOT:PSS with elastomers have been studied for stretchable electronics due to the decent stretchability and elasticity, but without successfully simultaneously achieving high electrical conductivity (few S/cm). On the other hand, ILs act not only as secondary dopants to significantly enhance the conductivity of PEDOT:PSS to reach the order of 1000 S/cm but also as plasticizers to improve the stretchability of PEDOT:PSS thin film carried by an elastomer substrate. Note that drop-cast thick films of PEDOT:PSS-IL composites are plastic, and not elastic, and fractured under strain below 25% (*ACS Appl. Mater. Interfaces* 2017, 9, 819; *Chem. Mater.* 2019, 31, 3519). Hence, up to now nobody has succeeded to create a highly conducting free-standing thick polymer layer that displays elasticity (and not only stretchability). That is the reason why nobody could yet demonstrate intrinsically stretchable TE modules to produce meaningful output power under mechanical stretching and this is the novelty of our manuscript. Although the output power can be significantly enhanced by altering module architectures from lateral to vertical configurations as discussed in our response to Reviewer #3's comment #8, the

development of high-performance stretchable TE materials and the first demonstration of intrinsically stretchable organic TE modules as well as their in-depth characterizations provide a number of important implications to the thermoelectric community including

- i) Importance of electromechanical properties of both TE legs and interconnects composing TE modules for stable energy harvesting under mechanical stretching (please refer our response to Reviewer #3's comment #5),
- ii) Independence of Seebeck coefficients from stretching conditions in the composites possessing excellent electromechanical properties (please refer our response to Reviewer #1's comment #2),
- iii) Strong correlation between Seebeck coefficients and electrical percolations in the composites supported by the suggested two DOS peaks model (please refer our response to Reviewer #3's comment #2),
- iv) Anisotropic behavior of the output power of stretchable TE modules under tensile strain in parallel and perpendicular directions (please refer our response to Reviewer #3's comment #6).

[Comment #2] What is the change in the Seebeck coefficients depending on the composition ratio of three substances, PEDOT:PSS, IL, and WPU?

[Response #2] To answer reviewer's comment, we measured Seebeck coefficient (S) of the composites over a full range of composition (i.e. different CP-WPU ratios, different contents of IL at optimized CP-WPU ratio, different types of IL) that we have characterized previously. The new results of S are now included along with the electrical and mechanical properties of the composites in Fig 2 & 3 in the revised manuscript. As shown in Fig. R5a, S of 21 $\mu\text{V}/\text{K}$ for the CP-TCM sample remains nearly the same with increasing the content of WPU (X) up to 85 (i.e. CP:WPU=(100- X): X =15:85). Note that the conductivity (σ_{dc}) of the composite decreases gradually with decreasing the content of CP up to this point. However, S starts to increase when $X > 90$ and reaches to $S=38 \mu\text{V}/\text{K}$ at $X=95$, while σ_{dc} decreases abruptly in this range. Strong correlation between S and σ_{dc} is also observed for the composites with different amount of IL (Fig. R5b). Here, the CP-WPU ratio is fixed, so a decrease in σ_{dc} with less amount of IL comes from a reduced extent of electrical percolation. On the other hand, the amount of IL itself didn't

affect S of CP-TCM samples without WPU (Fig R6a). Note that we measured S for all samples 3 or 4 days after the sample preparation due to the time dependence of S (Fig R6b) with the recovery of oxidation level over time (please refer our response to Reviewer #3's comment #4).

Figure R5. a) Electrical dc conductivity (σ_{dc}) and S of the composite with varying proportion of WPU (X) to CP (100–X). The composition ratio of the chosen IL (TCM) to CP was fixed at CP:TCM=15:25. b) σ_{dc} and S of the composite with varying content of TCM (Y). The proportion of WPU to CP was fixed at CP:WPU=15:85. Composition ratio, CP:TCM:WPU=15:Y:85.

Figure R6. a) S of CP films with varying content of TCM (Y). The relative content of CP is 15. 0.15 wt% of an ammonia solution was added with respect to the PEDOT:PSS dispersion as in the case for the composite with WPU. b) S of the optimized composite (CP:TCM:WPU=15:25:85, w/w) as a function of storage time in ambient conditions.

In order to explain this phenomenon, the increase in S with the reduced electrical percolation, we propose the following hypothesis. In good approximation, the transport of charge carriers in a hopping regime between localized states is governed by the density of state (DOS). The Fermi distribution dictates the filling of the DOS; which also determines a level of energy called the transport energy E^* (*J. Phys.: Condens. Matter* 1997, 9, 2699), dominating the percolation transport because of its highest hopping probability. The Seebeck coefficient (S) can be described as (*J. Appl. Phys.* **2003**, 93, 4653; *Phys. Rev. B* **2015**, 92,035201):

$$S \cong -\frac{E^* - E_F}{qT}$$

Where q is the charge of the carrier, T the temperature, E_F is the Fermi Level.

The density of percolation paths in PEDOT:PSS is strongly affected by the morphology of the film as known by the addition of (i) high boiling point solvents and ionic liquids that promotes the demixing of the excess of insulating PSS and promotes the creation of 3D percolation paths (*Adv. Electron. Mater.* 2015, 1, 1500017; *Adv. Mater.* 28, 8625 (2016)); or (ii) insulating polymers breaking the percolation paths (*Adv. Funct. Mater.* 2014, 24, 2957). Sometimes small amounts of those additives lead to a drastic change in transport properties. For instance, the same absorption spectrum is observed for films with an electrical conductivity varying by 3 orders of magnitude (*PNAS* 2018, 115, 11899), thus indicating that the concentration of PEDOT chains is barely affected, but their connectivity is much different and governed by microscopic morphology. Hence, we can thus speak about PEDOT chains with good connectivity and bad connectivity. PEDOT chains with good connectivity can form nanocrystals where short range order dominates the transport (blue domains in Fig. R7). While PEDOT chains with bad connectivity are in an amorphous phase without even local order (green domains). The electrostatic potential profile on those two different types of PEDOT chains is different (*Macromol. Rapid Commun.* 2018, 39, 1700533) and it is thus expected to affect their electronic energy levels. We propose that their DOS are thus also different. As a result, our hypothesis is to consider a DOS for PEDOT:PSS systems composed two DOS peaks. One DOS peak representing the electronic levels for the connected PEDOT chains and another DOS peak representing the electronic bipolaronic levels of the disconnected PEDOT chains. In this hypothesis, we thus propose that upon the removal of ionic liquid or the addition of insulating WPU polymer, the

ratio of connected vs. disconnected PEDOT chains decreases and the intensity ratio of those two DOS peaks differs, which leads to an increase of S as proposed recently for polymer blends (*Adv. Electron. Mater.* 2019, 5, 1800821).

Figure R7. Schematic drawings of different regimes of PEDOT connectivity (bottom) and the corresponding electronic energy structures (top).

This hypothesis is now included in the revised Supplementary Information. We revised our manuscript in accordance with new results and a hypothesis as shown below.

(on page 7) Up to this point, the S of the composite is nearly identical ($S \approx 22 \mu\text{V/K}$), but further loading of WPU increased S to $38 \mu\text{V/K}$ at WPU:CP=95:5.

(on page 8) As the electrical percolation diminishes with lower amounts of IL, the S increases inversely. Note that the amount of IL has no effect on the S of CP-TCM composites without

WPU (see Supplementary Fig. S1). A potential hypothesis to understand the correlation of S with percolation is proposed in Supplementary Information (Fig. S2).

(on page 19) Note that we measured S for all samples 3 or 4 days after the sample preparation due to the time dependence of S with the recovery of oxidation level over time (see Supplementary Fig. 1b).

[Comment #3] On line 206, the author referred to a paper that is not yet published.

[Response #3] We thank the reviewer's correction. Since this paper is still under consideration of publication, we removed this reference and explained the reason of adding ammonia in more detail in the revised manuscript as shown below.

(on page 10) Since we added a small amount of ammonia (CP:NH₃=8:1) in all composite dispersions to prevent acid-induced aggregation of anionic WPU by protonation from PSSH,

[Comment #4] On line 209, the author claims that the conductivity of composite decreased by dedoping is recovered via chemical equilibrium with the oxygen in air. However, no evidence or reference can be found to support the author's claim in the manuscript. Provide the appropriate evidences or refer relevant papers.

[Response #4] To provide evidences for dedoping of PEDOT by adding ammonia (NH₃) into the PEDOT:PSS dispersion and recovery of doping level in air after film formation, we measured absorbance spectra of CP-TCM films as a function of storage time. It is well-known that optical transitions observed in absorption spectra reflect electronic states governed by doping level (*J. Mater. Chem. C* 2014, 2, 1278). Doped PEDOT shows two main absorptions: a broad free carrier absorption in NIR region and a polaron absorption peak at around 900 nm. Note that these absorption features are not clear in the spectra of CP-TCM-WPU films due to the absorption of WPU (*J. Mater. Chem. A*, 2018,6, 9192). As shown in Fig. R8, the CP-TCM film coated from the

dispersion containing NH₃ (CP:NH₃=8:1) initially shows a pronounced polaron peak at around 900 nm and a reduced free carrier absorption in the NIR region as a result of partial dedoping of PEDOT occurring in the relatively basic dispersion compared with the pristine PEDOT:PSS dispersion (*J. Mater. Chem. C*, 2015, 3, 10616; *Int. J. Photoenergy*, vol. 2012, Article ID 598903). As the CP-TCM-NH₃ film was stored in air, the NIR absorption increased with a decrease in the polaron peak and the spectrum became identical with that of the CP-TCM film coated from the dispersion not containing NH₃ after 25 hours of storage in air. It has been proven that electrochemically reduced PEDOT is oxidized spontaneously upon oxygen exposure in the dry state (*J. Mater. Chem. A*, 2017, 5, 4404). In response to the reviewer's comment, we added this new data in the Supplementary Information as evidences and provided references in the revised manuscript as shown below.

(on page 10) the initial σ_{dc} was lower due to the partial dedoping of the PEDOT occurring in the relatively basic dispersion containing NH₃ (see Supplementary Fig. S7a,c)^{36,37}. The increase in σ_{dc} of the composite over time is attributed to the recovery of the doping level via spontaneous oxidation upon oxygen (air) exposure³⁸ as evidenced by the changes in absorption spectra (see Supplementary Fig. S7c).

(on page 8 in Supplementary) The CP-TCM film coated from the dispersion containing NH₃ initially shows a pronounced polaron peak at around 900 nm and a reduced free carrier absorption in the NIR region¹⁰ as a result of partial dedoping of PEDOT. As the CP-TCM-NH₃ film was stored in air, the NIR absorption increased with a decrease in the polaron peak and the spectrum became identical with that of the CP-TCM film coated from the dispersion not containing NH₃ after 25 hours of storage in air. Composition ratio, CP:TCM:NH₃=12:20:1.5 or CP:TCM=12:20, w/w.

Figure R8. Comparison of absorbance spectra of CP-TCM films over time. The films were stored in ambient air. Composition ratio, CP:TCM:NH₃=12:20:1.5 or CP:TCM=12:20, w/w.

[Comment #5] On line 318, are the Au-TiO₂ NWs stretchable? (If they are stretchable enough, please provide the proper evidence.) Resistance of the Au-TiO₂ NWs is significantly increased with an increase of tensile strain in Figure S15. I wonder the drop in the output power of the TE module in Figure 5f is closely related to increased resistance of the Au-TiO₂ NWs.

[Response #5] The Au-TiO₂ NWs embedded in PDMS substrates can be called “stretchable conductors” because this material is electrically conductive upon repeated tensile stretching above 100% owing to the retained connection between conductive NWs even under large deformations (see Fig. S18a in the revised Supplementary and ref. *Adv. Mater.* 2018, 1706520). However, the increase in R of Au-TiO₂ NWs when stretched is relatively large compared with the increase observed for the composite. So, the reviewer’s insight is correct, and we agree that it is important to address explicitly the main cause of R_{in} (total internal resistance of TE module) increase that proportionally reduces the output power of the TE module. Equivalent circuits drawn in Fig. R9 (Fig. S19 in the revised Supplementary Information) shows that R_{in} is the summation of resistances of a TE leg (R_{leg}), an Au-TiO₂ NW interconnect (R_{Au}), and a contact resistance between a leg and an interconnect (R_c). Comparing the changes in R_{in} and ΣR_{Au} upon

tensile stretching reveals that the R_{in} increase arises mainly from the increase in ΣR_{Au} , while the change in $\Sigma R_{leg} + \Sigma R_c$ estimated by subtracting ΣR_{Au} from R_{in} has minor contribution to the R_{in} increase. In accordance with our response to the next comment, we did the same analysis on the TE module stretched in the direction perpendicular to thermodiffusion. It is interesting to see that ΣR_{Au} increases upon perpendicular stretching even though it is expected to decrease slightly based on geometric changes (i.e. decrease in length and thickness, increase in width, while $R \propto \text{length}/(\text{width} \times \text{thickness})$), presumably due to the reduced density of connections between conductive NWs. A slight increase in $\Sigma R_{leg} + \Sigma R_c$ can be understood in the same manner for CP nanofibrils. This analysis is included in the revised Supplementary Information with the revision of main text as shown in our response to Reviewer #3's comment #6.

Figure R9. a,b) Schematics of TE modules characterized in Fig. 6 under tensile strain in the direction (a) parallel with and (b) perpendicular to thermodiffusion. Equivalent circuits to calculate the internal resistance of TE modules (R_{in}) were drawn below. $R_{in} = \sum R_{leg} + \sum R_{Au} + \sum R_c$, where R_{leg} , R_{Au} , and R_c are the resistances of a TE leg, an Au-TiO₂ NW interconnect, and a contact between a leg and an interconnect, respectively. c,d) Contributions of $\sum R_{Au}$ and $\sum R_{leg} + \sum R_c$ to R_{in} . $\sum R_{Au}$ was estimated by multiplying R of a single interconnect measured under different

tensile strain and the number of interconnects in each module (i.e. 9 for (c) and 4 for (d)). R_{Au} vs. strain (ε) in (c) was obtained on a motorized linear stage with gold-coated 4-point contact pads, while R_{Au} vs. ε in (d) was measured in the same way as for the TE module by the 2-point probe method. $\Sigma R_{leg} + \Sigma R_c$ vs. ε was estimated by subtracting ΣR_{Au} from R_{in} . Comparing the changes in R_{in} and ΣR_{Au} reveals that the R_{in} increase arose mainly from the increase in ΣR_{Au} , upon both parallel and perpendicular stretching, while the change in $\Sigma R_{leg} + \Sigma R_c$ has minor contribution to the R_{in} increase. Despite a reduced length of interconnects upon perpendicular stretching, the ΣR_{Au} increase can be caused by the reduced density of connections between conductive NWs.

[Comment #6] In page 30, the authors evaluated the TE properties of the stretchable TE module in only one direction. Since the active part of the TE module shown in Figure 5 has a long rectangular shape with anisotropy (not a square), they should be characterized in both perpendicular and parallel direction to elongation.

[Response #6] In response to Reviewer #3's comment, we characterized the performance of the TE module upon tensile stretching in the direction perpendicular to thermodiffusion as shown in Fig. R10 (Fig. 6 in the revised manuscript). As we have already discussed in our response to Reviewer #3's comment #5, R_{in} was increased even by perpendicular stretching, but the increase was much less than in the case of parallel stretching (Fig. R10b, e) so that 83% of the initial $P_{out-max}$ is maintained under 40% strain (Fig. R10f). Note that we couldn't help but reduce the later size of the module, i.e. the number of TE legs in the module, for a practical reason that the Peltier elements equipped in our set-up have a limited size and can't cover 10-legs modules when stretched to 40% in a perpendicular direction. Accordingly, we revised our manuscript as shown below.

(on page 15) The tensile loading up to 40% in both parallel and perpendicular directions does not affect the V_{OC} of the stretchable module (Fig. 6e,h). However, the R_{in} increases gradually with increasing strain due to the strain-dependent resistances of the composite legs and the interconnects (see Supplementary Fig. S18). The analysis of the strain-dependent resistances with equivalent circuits proves that the R_{in} increases arose mainly from increased resistance of the

Au-TiO₂ NW interconnects for both stretching directions, while the R_{in} increase was much less pronounced for perpendicular strain (Fig. 6e,h and Supplementary Fig. S19). According to the relation $P_{out} \propto 1/R_{in}$, the $P_{out-max}$ gradually decreases with increasing strain, with 48% and 83% of the initial $P_{out-max}$ maintained for 40% parallel and perpendicular strain, respectively (Fig. 6f, 6i).

Figure R10. a-c) TE module stretched in the direction parallel with thermodiffusion. $T_1=23$ °C, $T_2=53$ °C. (b) V_{OC} and R_{in} vs. tensile strain (ϵ) applied to a module for $\Delta T=30$ K. (c) P_{out} vs. R_{load} under different strains for $\Delta T=30$ K. d-f) TE module stretched in the direction perpendicular to thermodiffusion and corresponding V_{OC} and R_{in} vs. ϵ and P_{out} vs. R_{load} .

[Comment #7] On line 650, the elasticity of TE module with the composite is different from that of the pristine active material because the TE module includes both substrate and electrode. Is there no effect of the substrate and the electrode on the performance of the active layer? Also, direction of stretching should be indicated with an arrow in Figure 5 for the reader's understanding.

[Response #7] Fig. R11 (Fig. S18b in the revised Supplementary Information) shows that the electromechanical property of the free-standing composite is different from the composite supported on a PDMS substrate. As the effect of supporting substrates is also shown for the electromechanical property under cyclic stretching (see Fig. R1 and our response to Reviewer #1's comment #2), we assume that different electromechanical properties originate from different mechanical deformations between the composite and a PDMS substrate since the composite ($\sim 40 \mu\text{m}$) supported on a PDMS substrate conforms to the substrate ($>200 \mu\text{m}$) when deformed. Nonetheless, the difference is not significant under tensile strain up to 40%. Meanwhile, embedding Au-TiO₂ NWs ($<5 \mu\text{m}$) in a PDMS substrate has a negligible effect on the mechanical property of the substrate as proven in the reference (*Adv. Mater.* 2018, 1706520), so neither on that of the composite supported on a PDMS. To clarify this, we added a new result in the revised Supplementary Information as shown below. Also, as suggested by Reviewer #3, we indicated the direction of stretching TE modules as well as the schematics of geometrical changes in TE modules in Fig. 6 in the revised manuscript.

Figure R11. Resistance of a TE leg (i.e. the optimized composite film supported on a PDMS substrate, gray line) as a function of tensile strain with the comparison to the free-standing optimized composite film (data from Fig. 3b, blue line). Different electromechanical properties

with and without PDMS support can originate from different mechanical deformations between the composite and a PDMS substrate since the composite ($\sim 40 \mu\text{m}$) supported on a PDMS substrate conforms to the substrate ($>200 \mu\text{m}$) when deformed. Nonetheless, the difference is not significant under tensile strain up to 40%.

[Comment #8] On line 305, for wearable and stretchable TE modules, vertical configuration is more suitable than parallel one due to its easy build-up of the temperature gradient and low electrical resistance. If the PEDOT:PSS-IL-WPU composite forms the enough thickness, vertical TE module can be prepared by using only p-type TE legs. Is the thickness of the PEDOT:PSS-IL-WPU films ($\sim 40 \mu\text{m}$) sufficient for the TE legs of vertical configuration? If it is impossible, how can the thickness be increased or controlled? How can the electrical conductivity and mechanical properties be optimized depending on thickness?

[Response #8] As we have also discussed briefly in the section following after line 305, we strongly agree with Reviewer #3 that vertical configuration is more suitable for wearable TE modules, especially to maximize the output power harvested from our body heat due to its geometry that may reduce R_{in} (i.e. enhance P_{out}) by 3-4 orders of magnitude compared with lateral configuration. The current thickness of our composite ($\sim 40 \mu\text{m}$) can already be used as the TE legs of vertical configuration in case of micro-TE generators (*J. Microelectromech. S.* vol. 27, no. 1, 1, 2018). But, the thickness of our composite can be increased simply by increasing the drop-casting volume of composite inks. Nonetheless, as we have discussed regarding Reviewer #3's comment #5, not only stretchable TE legs but also stretchable interconnects are important elements to build stretchable TE modules, and the demands on both materials and processes for interconnects would be very steep to build vertical TE modules without comparable n -type TE materials. Therefore, we think that this issue is out of scope for this work but is important and should be considered for the next project.

REVIEWERS' COMMENTS (After revision):

Reviewer #1 (Remarks to the Author):

I appreciate the effort the authors expended into fully addressing the reviewer comments. I believe this manuscript not only represents a significant advancement in the field of flexible thermoelectrics, but also contains sufficient detail and corollary data to motivate and guide the research efforts of the broad flexible electronics community and is therefore suitable for publication in its current form.

Reviewer #3 (Remarks to the Author):

In this revised manuscript, experimental results are systematically demonstrated to address a relationship between Seebeck coefficient and electrical conductivity of elastic conducting polymers with IL and WPU. Also, new investigation is suggested for the analysis on anisotropy of a thermoelectric module for stretchable electronics and effect of Au-TiO₂ NW electrodes on thermoelectric performance with relevant discussions. This revised manuscript is significantly improved and I believe this manuscript is sufficient for publication in Nature Communication.